# Feasibility of Using *Tenebrio molitor* Larvae as an Alternative Protein Source

**DOI:** 10.3390/foods14234068

**Published:** 2025-11-27

**Authors:** Rubén Agregán, Noemí Echegaray, Laura Moraga-Babiano, Mirian Pateiro, José M. Lorenzo

**Affiliations:** 1Fundación Centro Tecnolóxico da Carne, Adva. de Galicia n° 4, Parque Tecnolóxico de Galicia, San Cibrao das Viñas, 32900 Ourense, Spain; noemiechegaray@ceteca.net (N.E.); laura.moraga@ceteca.net (L.M.-B.); jmlorenzo@ceteca.net (J.M.L.); 2Área de Tecnoloxía dos Alimentos, Facultade de Ciencias, Universidade de Vigo, 32004 Ourense, Spain

**Keywords:** sustainability, mealworm, insect farming, insect protein, mealworm composition, mealworm health risks, mealworm profitability

## Abstract

Mealworm farming is gaining interest as a possible solution to the oversaturated meat supply chain, as an alternative source of protein. This is a more environmentally friendly activity that requires fewer inputs for production compared to meat. This review discusses the feasibility of mealworms as an ingredient for the production of novel foods, investigating crucial aspects, such as nutrition, technological capability, food safety, and consumer acceptance, among others. *Tenebrio molitor* larvae can be nutritionally comparable to meat, as they provide high-quality protein and other essential nutrients. Although the omega-6/omega-3 ratio exceeds the recommended limit (<5), certain strategies during larval breeding, including feeding, and cooking, may significantly reduce this gap. The use of mealworm flour in the food industry can provide apparently healthy, safe matrices with high protein content. However, inclusions above 10% often lead to technological and sensory deficiencies. Further experimentation is required to overcome these issues, which negatively impact consumer acceptance, and to promote social behavioral strategies to attract consumers toward insects. On the other hand, regulatory policies might play a crucial role in supporting this business, which is predicted to grow as technology develops and this activity aligns with a circular economy.

## 1. Introduction

The excessive consumption of meat in recent decades and the projected increase by 2050, especially in low and middle-income countries, is depleting natural resources, since meat has significantly larger environmental and climate footprints than the production of plant-based foods [1]. In this regard, several initiatives focused on reducing meat protein dependence have recently been applied, such as the exploitation of meat processing waste, the use of plant-derived meat substitutes, in vitro cultured meat, and insect farming [2].

Insects have been assessed as an alternative protein source to meet the increasing protein demand worldwide, also promoting the development of sustainable food systems [3]. Therefore, insect farming seems a promising, sustainable activity that might provide food security in low-income communities. However, before implementing large-scale production, it is necessary to deeply investigate the nutritional characteristics of these animals, as well as their food safety and profitability.

There are currently six commissions implementing regulations authorizing four food business operators in the European Union (EU) to market products made of four edible insect species: the yellow mealworm (*Tenebrio molitor*), the lesser mealworm (*Alphitobius diaperinus*), the house cricket (*Acheta domesticus*), and the migratory locust (*Locusta migratoria*) [4].

The larva of *T. molitor*, commonly known as mealworm (from this point in the text, we will refer to this insect as mealworm or *T. molitor* alternately), is a promising and interesting alternative to meat protein due to its chemical composition, as it is a rich source of nutrients. Thus, the mealworm stands out for its high protein content, with a balanced amino acid profile, healthy fatty acids (FAs), vitamins, and minerals [5]. This insect species can be reared with relatively low resource input, saving natural resources, such as water and land, compared to meat livestock [6,7]. In addition, mealworm farming releases fewer greenhouse gases (GHGs) and produces lower emissions of NH_4_ (ammonium), NH_3_ (ammonia), and CH_4_ (methane) than livestock breeding, impacting less on the environment [8]. For all these reasons, mealworms are being widely investigated by the scientific community as a realistic solution to the sustainability problems derived from excessive meat consumption worldwide.

In this review, the authors want to provide an overview of mealworms as a potential protein-rich ingredient to replace other traditional protein sources, such as meat. To this end, several key points are discussed that determine the introduction of mealworms as an efficient ingredient to provide technologically well-formed food products with consumer acceptance. In addition, the potential hazards of using the *T. molitor* larvae as a food ingredient, including allergies, possible transmission of viruses and prions, and the presence of heavy metals, among others, are discussed in depth. The economic outlook and profitability of using this insect species in the food market are also analyzed.

## 2. Insects as an Alternative Source of Protein for Future Sustainability

Meat is the primary source of dietary proteins for many populations around the world, contributing high-quality nutrients for a balanced diet. Up to 22% of the total product weight may be represented by proteins [9]. However, the recent increase in meat consumption worldwide, especially in the last two decades, threatens the sustainability of ecosystems, leading future generations to face potentially dire environmental consequences [10]. The current livestock supply system is the largest producer of CH_4_ worldwide and is responsible for 14.5% of global anthropogenic GHG emissions, also promoting biodiversity loss and extensive water use [11,12]. In addition, it is estimated that the demand for meat will increase by around 60–70% by 2050, according to the purchasing dynamics observed in developed countries and those in economic transition [12,13], thus aggravating the scenario.

The above challenges must be addressed by exploring sustainable consumption models in order to alleviate pressure on natural resources while ensuring food security. Research into alternative protein sources has rapidly emerged as an interesting way to meet the requirements for this nutrient and counter the global concerns mentioned above [14]. In this regard, several promising protein matrices, distinct from meat, have been investigated recently to serve as alternatives in food product development. These are in vitro cultured meat as an artificial replication of its natural counterpart, plant and insect proteins, and proteins from algae and yeasts [10]. As far as insects are concerned, their dietary inclusion has positively impacted the food sector and consumers, providing high levels of proteins, vitamins, and minerals and exhibiting potential positive health effects in the body, such as the prevention of chronic diseases (e.g., diabetes, cancer, and cardiovascular disease (CVD)). In addition, insects show a remarkably high feed conversion, producing high amounts of biomass potentially usable for human nutrition [15,16]. In addition, insect farming requires considerably less land and water than livestock rearing. Van Huis [17] reported a difference of more than 200 m^2^ between mealworm production and beef production (<50 vs. >250 m^2^) to obtain 1 kg of edible protein. Similarly, the water requirement to produce 1 kg of both biological materials differed by more than 70 L (<30 vs. >100 L for mealworms and beef, respectively). In this way, the carbon footprint could be dramatically reduced by more than 150 CO_2_-equivalent [17].

Therefore, insect farming might positively contribute to the sustainability of meat supplies for the coming future (Figure 1). Applying a circular economy model would convert food waste into high-quality proteins that are potentially usable to replace fishmeal and soybean meal in livestock diets [18,19]. This approach ensures food security by providing alternative meat proteins, a different type of feed for livestock, improved food production, and the release of farmland. This scenario might promote new profitable opportunities for small farmers and contribute to rural development [20].

In addition to all the above, exploring both consumer appreciation for edible insects and their purchasing intentions will be determining factors for the effective incorporation of insect-based protein into the global food system. In addition, ensuring food safety, as well as analyzing the profitability and scalability of the business model, will also play a key role in marketing and stabilization of this type of product in the food market.

## 3. Possibilities of Mealworms as a Sustainable Meat Protein Replacement

The mealworm is the larval form of the yellow mealworm beetle, a widespread type of insect that can usually be found infesting meals and other stored agricultural products, including grains and related starch products, such as bran and pasta. This worm is included in the order Coleoptera and in the family Tenebrionidae [21]. *T. molitor* is believed to be originally from the Mediterranean basin, although it can currently be found distributed throughout the world due to colonization and trade [22].

The mealworm is a holometabolous insect. This means that it undergoes a complete metamorphosis, passing through all the phases until adulthood. These are egg, larva, pupa, and adult insect. The larva is well sclerotized, has an elongated cylindrical shape, measures approximately 25 mm in length, and weighs more than 200 mg in the final stage of development. The mealworm’s body has six legs located behind the head and two short appendages at the tips of the abdomen. The color is initially white and progressively turns to a yellowish-brown tone. The beetle’s larval stage lasts from 3 or 6 months to 2 years. This larval stage consists of four to five developmental periods, the duration of which depends on temperature, food availability, and larval density in the breeding environment. Optimal developmental conditions seem to be around 60–75% of relative humidity and a temperature of 25–28 °C. Diet was also found to be critical for optimal larval growth, with 5 to 10% of yeast and 80 to 85% of carbohydrates, along with the addition of B vitamins, providing the best conditions to achieve a good yield. Long periods occur when survival conditions are unfavorable for pupae, usually in winter. Then, the larva transforms into a pupa during the spring season, adopting a “C” shape. This new phase is considerably shorter, lasting between 7 and 48 days, depending on the temperature. Finally, in early summer, the lifecycle of *T. molitor* ends, and it develops a hard, shiny, blackish-brown shell with a body length between 1 and 1.4 cm and a life expectancy of 37–63 days [23,24,25].

### 3.1. Nutritional Attributes of Mealworms

#### 3.1.1. Protein Content

Analysis of the mealworm’s body showed adequate amounts of protein, fat, and fiber, as well as vitamins and minerals (Table 1). In addition, some specific chemical compounds, such as chitin and biopeptides, were also found [26]. The main chemical compound reported is protein, with values close to 50% in dry larvae (DL) (Table 1). However, higher percentages are possible (>70% DL) [27]. Mealworm protein amounts can be comparable to those of meat. Thus, while the protein in larvae can reach about 20% of fresh weight (FW) (Figure 2), some meat cuts, such as pork shoulder, beef sirloin, and chicken breast, barely reach this percentage (16.9, 20.1, and 21.5% of FW, respectively) [28]. This high amount of protein is constituted by a highly balanced amino acid profile, especially highlighting the presence of leucine and isoleucine (≥60 mg/g DL), both of which are essential amino acids (EAAs). Lysine can also be found at high levels (>50 mg/g DL) [26]. Ravzanaadii et al. [29] reported values of up to 71, 82, and 84 mg/g DL for the EAAs histidine, phenylalanine, and threonine, respectively. Meat cuts, such as pork shoulder and beef sirloin, are also rich in leucine and lysine. Orkusz [28] found similar leucine values when comparing the previous cuts (14.3 and 16.8 mg/g of edible portions for pork shoulder and beef sirloin, respectively) with *T. molitor* larvae (14 mg/g of the edible portion).

Non-essential amino acids (NEAAs) also contribute to the formation of mealworm proteins, especially glutamic acid, which can reach values above 100 mg/g DL, but also aspartic acid, proline, and tyrosine [26]. This last amino acid was found at higher levels (13.7 mg/g of the edible portion) than in pork shoulder (6.2 mg/g of the edible portion), beef sirloin (7.5 mg/g of the edible portion), and white meat cuts, such as chicken and turkey breast (7.4 and 6.2 mg/g of the edible portion, respectively) [28].

Scientific literature on the digestibility of mealworm protein is still scarce. Poeaert et al. [30] reported values between 84 and 92% when assessing crude protein and between 84 and 94% after heating. In addition, analysis based on the protein digestibility corrected amino acid score (PDCAAS) revealed adequate values to cover the requirements of the human diet (69–84%). However, certain limitations were reported when using this method, and others, such as the digestible indispensable amino acid score (DIAAS), recently considered by the Food and Agriculture Organization (FAO), as it is a more accurate method to measure protein digestibility in food matrices. This method considers the individual digestibility of amino acids in the ileal region of the intestine, using a growing pig as the preferred model, rather than the rat used in PDCAAS, and avoids mathematical truncation in the final score [31]. In contrast, the PDCAAS method assumes equal digestibility for crude proteins and all individual amino acids. Using this method, the proteins generated by the metabolism of bacteria inhabiting the gastrointestinal tract are considered in the calculation, leaving aside those of dietary origin used by these bacteria. Finally, protein sources that provide more EAAs than required cannot be compared or ranked [32]. Using DIAAS to estimate protein digestibility via an in vitro procedure, Lampová et al. [33] found that culinary treatments, such as boiling, roasting, drying, and microwave heating, could improve mealworm digestibility. Specifically, drying was the most effective method for enhancing protein digestibility, increasing by more than 5% (from 115.4 to >120%). Conversely, Hammer et al. [34] observed that a bleaching process in mealworms facilitated a better in vitro DIAAS score (85%) than other procedures, including freeze-drying and pulverization. However, it was still far from the score reported for chicken meat (113%). Therefore, there appears to be a direct relationship between the use of heat in mealworm processing and the digestibility of its protein. Despite this fact, mealworm protein could be classified as good quality for human consumption, according to the FAO’s recommended classification based on the DIAAS values for each matrix. Thus, proteins with a score between 75 and 99% are considered high quality, and those above 100% are considered excellent [31].

#### 3.1.2. Other Nutritional Compounds of Mealworms

##### Fat

Fat is the second most representative component of the mealworm’s body after protein, representing more than 30% DL (Table 1). Oleic (C18:1*n*–9c) and linoleic (C18:2*n*–6c) acids (about 39 and 35% of total FAs, respectively) (Figure 2) are the most abundant FAs reported in this fat. Oleic acid was found in the larvae at similar levels to pork shoulder (37.5% of total FAs) and pork belly (36–40% of total FAs), while higher values were reported in beef (50% of total FAs) [35,36,37]. The content of linoleic acid is considerably higher than in meat, accounting for about 35% of total FAs (Figure 2). On the contrary, meats, such as beef, lamb, and goat, barely exceed 10% of total FAs. In the case of pork and horse meat, slightly higher values are observed, sometimes exceeding 20% of total FAs [38]. Linoleic acid is part of the omega-6 polyunsaturated FA (PUFA) series, which has been suggested to promote a protective effect on the cardiovascular system when consumed in adequate amounts [39].

Palmitic acid is the third most important FA in the mealworm, representing 10–20% of total FAs (Table 1). This proportion is noticeably higher for meats, such as pork and beef, with values between 20 and 30% of total FAs [38]. Finally, it is worth mentioning the presence of linolenic acid (C18:3*n*–3) in the larvae, although in a remarkably lower quantity (2% of total FAs) than oleic and linoleic acids (Table 1). This FA belongs to the omega-3 PUFA series, recognized for its potential beneficial effects on cardiovascular health [40]. However, these positive body actions might be mitigated by the excessively high omega-6/omega-3 (*n*6/*n*3) in the larvae. Jajić et al. [27] reported a value of around 12 for this ratio, far exceeding the maximum value estimated as healthy by the FAO (<5) [41]. This figure is consistent with that found in another study (20.8), which was four times higher than the recommended limit [42]. This indicates a significant and dangerous imbalance between omega-3 and omega-6 FAs, which could lead to obesity and the pathogenesis of many diseases [23].

Recent studies suggest that the type of mealworm diet may influence the quality of fat, altering the FA profile. Thus, increased amounts of omega-3 PUFAs might positively change the accumulation of these compounds in the edible tissues of the larva [43,44]. This finding could considerably reduce the high *n*6/*n*3 ratio, making mealworm fat healthier. Dragojlovic et al. [45] reported this fact when administering different rearing substrates to *T. molitor* larvae. The authors highlighted the use of flaxseed, an abundant source of linolenic acid (C18:3*n*–3), to increase the *n*6/*n*3 ratio. In general, feeding mealworms with a mixture of cabbage, carrot, and flaxseed (1:1:1, *w*/*w*) increased the omega-3 FA. A combination of feeding and different harvest times can achieve really low values for the *n*6/*n*3 ratio (1.6), as these authors demonstrated. Interestingly, a change in diet, passing from wheat bran to the aforementioned vegetable sources, hardly changes the digestive capacity of the larvae. Regarding the improvement in protein digestibility by heating the mealworm discussed in the previous section, Mancini et al. [46] suggested that the *n*6/*n*3 ratio is also significantly affected when intense high-temperature treatment is applied to the larvae. Thus, oven cooking at 70 °C for 30 min can reduce this ratio from 5.5 to 1.5. However, cooking methods should be chosen carefully, as the same authors reported an increase in the ratio to almost 80 when deep frying or steaming for 20 min, due to the higher proportion of the omega-6 linoleic FA. This event is related to oxidation processes during heating, affecting unsaturated FAs. These processes could be modulated, according to the authors.

As can be observed, attempts are being made to produce *T. molitor* larvae with improved *n*6/*n*3 ratios. Despite these achievements, further research is needed to ensure not only a good FA profile in this insect, but also a balanced overall nutritional profile.

**Figure 2 foods-14-04068-f002:**
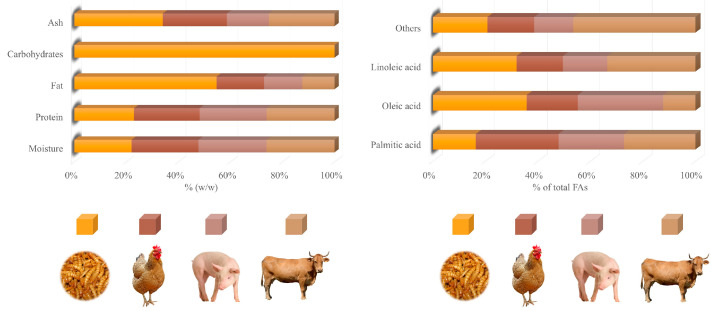
Comparison of different meats (chicken, pork, and beef) and mealworms on proximate composition and major fatty acids. Data are expressed as a percentage of fresh weight and a percentage of total fatty acids analyzed. FA: fatty acid. Data from Bhawana et al. [47]; Chernukha et al. [48]; Costa et al. [42]; Dal Bosco et al. [49]; Eom et al. [50]; Liza et al. [51]; Lukic et al. [52].

##### Carbohydrates

The amount of carbohydrates in mealworms is lower than that of protein and fat. However, studies have not focused much on this macronutrient so far. Kim et al. [53] reported that only 9.6% of the larval body, expressed as dry weight (DW), contained carbohydrates. Son et al. [54] found a similar value (11.5%) for this macronutrient. Interestingly, approximately 30% of the carbohydrates in mealworms are soluble sugars, particularly fructose. Considering that the threshold for a sugary taste is 0.125% [55], the use of mealworms in cooking might provide a slight sugary sensation, since free sugars represent 0.120% DL [54]. 

Almost half of the carbohydrates in mealworms, around 5% of the total dry body, are represented by chitin [54,56], which is the main dietary fiber in these insects [26]. Its consumption is associated with several health properties, such as the modulation of intestinal microbiota, the improvement of lipid metabolic disorders, anti-obesity effects, cardiovascular protection, and anti-cancer activities against colon and pre-menopausal breast cancers [57,58,59]. Nevertheless, reported concerns about the allergenicity of chitin in crustaceans and mollusks, as it is the primary component of their exoskeleton [60], might represent a significant obstacle to the consumption of mealworms by sensitive people. Chitin molecules can be involved in the production of allergen-specific IgE antibodies, causing immediate hypersensitivity reactions [61]. Another problem related to chitin intake is the possibility of protein encapsulation in the small intestine, preventing absorption. This event could reduce the digestibility of mealworm proteins, accelerating gastrointestinal transit [62]. In this sense, choosing insects with low chitin content might improve protein digestibility and nutritional value. However, some potentially harmless treatments for the consumer have recently been tested in order to attempt to remove chitin molecules from the body of *T. molitor* or to infer damage to its structure. Bogusz et al. [63] found structural changes in this polysaccharide when the larvae underwent a freeze-drying process. The ice crystals formed might damage the chitin molecule, impacting the nitrogen content. In another study, the use of ultrasonication and pulsed electric fields (PEFs) resulted in structural modifications of chitin, allowing the enzyme chitinase to act and produce chitooligosaccharides, which are interesting hydrolysis products with biofunctional properties [64]. Despite the concern about its allergenicity in sensitive individuals, mealworm chitin is not yet considered a major concern by the European Commission based on the research data collected to date. Still, they advocate for warnings on the labels of products that contain it to highlight the possible risk of allergic reactions [65]. On the other hand, chitin is an important dietary fiber, as mentioned above, and removing it from the larvae would decrease the final quality of the product. Therefore, this issue should be critically assessed before considering commercialization.

##### Vitamins and Minerals

The micronutrient profile of mealworms is characterized by the presence of relevant amounts of vitamins, including B-complex vitamins, such as vitamin B3 (niacin), vitamin B7 (biotin), and vitamin B9 (folic acid), all of which are between 100 and 1000 mg/kg DL (105.9, 948.7, and 300 mg/kg, respectively) [66]. Meat is also a natural source of vitamins. Pork and chicken are rich in vitamin B5 (pantothenic acid), and vitamin B6 (pyridoxine) can be found in beef, pork, and turkey, and all of these meats, along with sheep, are sources of vitamin B12 (cobalamin) [67]. The mealworm also contains appreciable amounts of vitamin B12, with a concentration of 1 µg/100 g DL [68]. In contrast, other vitamins, such as A (retinol) and D3 (cholecalciferol), are barely present [69,70]. Likewise, lean meat is not rich in these vitamins or vitamin E (tocopherol). On the other hand, beef and chicken livers are important sources of vitamin A [67]. Unlike meat, mealworms are a natural source of tocopherol, which is widely recognized for its antioxidant properties [71,72,73]. A concentration of 144.3 mg tocopherol/1000 g of larval oil has been reported in the literature. Vitamin C (ascorbic acid) is another micronutrient that differentiates mealworms from meat. While meat is not a natural source of this compound, the mealworm can provide 61.5 mg/kg DL [66,67].

*T. molitor* larvae contain important minerals for human nutrition, such as calcium, iron, and sodium (Table 1). Magnesium and phosphorus are also found in large quantities (>1000 mg/100 g DL), as well as potassium, with values around 1000 mg/100 g DL [26,42,74]. Meat is an excellent source of the latter mineral, with about 350 mg/100 g in the lean meat of pork, chicken, and turkey. Beef and chicken livers are also good sources of potassium, showing levels similar to those of lean meats (300–350 mg/100 g of liver) [67]. The iron content in the mealworm’s body is even higher than in several meats, such as beef, pork, and chicken. According to the data shown in Table 1, and assuming a theoretical moisture content of 60% in larvae, the level of this mineral could be around 3.5 mg/100 g of FW, compared to 2.7 and 0.7 mg/100 g in the lean meat of beef and pork, respectively [67].

Interestingly, the sodium content in mealworms is quite low compared to other minerals, such as phosphorus and potassium (40–400 vs. >1000 mg/100 g DL). Similarly, meat does not have a high content of this element either, presenting values below 100 mg/100 g of lean meat [67].

Based on these findings, *T. molitor* larvae could be considered a potential source of protein, contributing a full spectrum of macro and micronutrients. In this sense, mealworms have been catalogued as potential meat substitutes according to international nutritional guidelines [72].

**Table 1 foods-14-04068-t001:** Nutritional composition of mealworms. Concentrations are expressed as dry larvae (DL).

**Proximal composition (g/100 g)**	Jajić et al. [27]	Costa et al. [42]	Hong et al. [26]	Kim et al. [53]	Ravzanaadii et al. [29]
Protein	38.9–71.2	45.6	46.07	49.8	46.4
Lipids	6.1–45.2	34.5	19.1–36.1	37.1	32.7
Carbohydrates	–	–	–	9.6	–
Fiber	6.3–10.4	–	4–22.4	–	4.6
Ash	3.5–7.5	4.1	2.7–4.2	3.5	2.9
**AAs (mg/g)**	Adámková et al. [75]	Costa et al. [42]	Hong et al. [26]	Wu et al. [74]	Ravzanaadii et al. [29]
**EAAs**					
Histidine (His)	13.9–17.4	13.2	8.4–28	8.4	71
Isoleucine (Ile)	19.8–25.1	14.6	13.1–64.8	13.1	16.5
Leucine (Leu)	33–39.2	34.6	22.1–62.1	22.1	15.8
Lysine (Lys)	23.4–26.6	11.9	15.8–53.1	15.8	13.5
Methionine (Met)	4–7.3	–	5.2–13.4	6.01	31
Phenylalanine (Phe)	14.9–18.2	15.6	13.1–32	13.1	82
Threonine (Thr)	18.5–22.3	25	12.7–33.1	12.7	84
Tryptophan (Try)	–	–	0.2–3	3	–
Valine (Val)	28.6–36.3	22.4	18.9–44.6	18.9	11.3
**NEAAs**					
Alanine (Ala)	31.9–42.5	46.0	24.8–67	24.8	17.1
Arginine (Arg)	21.8–28.4	20.8	18.9–44.2	18.9	11.3
Aspartic acid (Asp)	40.9–47.6	44.4	15.4–65.2	15.4	16.7
Cysteine (Cys)	4.1–6.7	–	5.2–11.9	11.9	2.4
Glutamic acid (Glu)	44.1–63.3	57	39.2–103.2	39.2	26.4
Glycine (Gly)	24–31.4	30.1	17.1–43.8	17.1	11.2
Proline (Pro)	32.1–43.5	43.7	20–55.2	20.01	14
Serine (Ser)	17.8–21.2	26.5	13.6–38.2	13.61	9.7
Tyrosine (Tyr)	22.6–27.3	30.6	21.5–63.2	21.46	16.1
**FAs (% of the total FAs)**	Jajić et al. [27]	Costa et al. [42]	Hong et al. [26]	Wu et al. [74]	Ravzanaadii et al. [29]
**SFAs**					
Myristic acid (C14:0)	0.16	4	2.12–5.21	2.12	3.05
Pentadecanoic acid (C15:0)	–	–	0.06–7.1	0.2	–
Palmitic acid (C16:0)	13–19.7	15.3	9.3–17.2	17.2	16.7
Stearic acid (C18:0)	1–2.1	2.7	0.3–3.1	0.7	2.5
**MUFAs**					
Palmitoleic acid (C16:1*n*–7)	–	2.8	2.1	1.9	2.7
Oleic acid (C18:1*n*–9)	16.1–35.3	37.8	40.8–49.7	43.8	43.2
**PUFAs**					
Linoleic acid (C18:2*n*–6)	33.3–55.3	33.2	29.4–35.6	29.4	30.2
Linolenic acid (C18:3*n*–3)	1.96–4.28	1.5	0.35–2.27	2.27	1.36
Eicosapentaenoic acid (EPA; C20:5*n*–3)	0.04–1.3	–	–	–	3.1
Docosahexaenoic acid (DHA; C22:6*n*–3)	–	–	–	–	–
**Vitamins**	Rumpold and Schluter [66]				
Retinol (vit. A) (µg/100 g)	–	–	–	–	–
Cholecalciferol (vit. D3) (µg/100 g)	–	–	–	–	–
Tocopherol (vit. E) (mg/kg)	–	–	–	–	–
Ascorbic acid (vit. C) (mg/kg)	61.5	–	–	–	–
Thiamin (vit. B1) (µg/100 g)	310	–	–	–	–
Riboflavin (vit. B2) (mg/kg)	4.1	–	–	–	–
Niacin (vit. B3) (mg/kg)	105.9	–	–	–	–
Pantothenic acid (vit. B5) (mg/kg)	37.2	–	–	–	–
Pyridoxine (vit. B6) hydrochloride	–	–	–	–	–
Biotin (vit. B7) (mg/kg)	948.7	–	–	–	–
Folic acid (vit. B9) (mg/kg)	300	–	–	–	–
Cobalamin (vit B12) (mg/kg)	–	–	–	–	–
**Minerals**	Costa et al. [42]	Hong et al. [26]	Wu et al. [74]	Kim et al. [53]	Rumpold and Schluter [66]
Calcium (Ca) (mg/100 g)	–	40–380	31	34.9	47.2
Copper (Cu) (mg/kg)	7.8	12.3–20	20.2	11.4	16.4
Iron (Fe) (mg/kg)	67.6	63–100	184.2	62.8	55.1
Magnesium (Mg) (mg/100 g)	282.3	200–1630	233.3	137.6	221.5
Manganese (Mn) (mg/kg)	11.6	–	18.9	7	9.2
Phosphorus (P) (mg/100 g)	797	700–1040	–	567.7	697.4
Potassium (K) (mg/100 g)	800.7	740–950	891.4	–	761.5
Sodium (Na) (mg/100 g)	206.6	110–360	43.7	–	125.4
Zinc (Zn) (mg/kg)	96.5	102–117.4	98.6	98.7	114.1

AA: amino acid: EAA: essential amino acid; NEAA: non-essential amino acid; FA: fatty acid; SFA: saturated fatty acid; MUFA: monounsaturated fatty acid; PUFA: polyunsaturated fatty acids.

### 3.2. Suitability of Recent Trends in Mealworm Farming for Production Yield

The feed conversion ratio (FCR) (feed consumed/weight gained) is a fundamental characteristic of every living organism that indicates its capacity for growth and the production of edible tissue; in other words, the efficiency in converting food into body weight. In this sense, insects can be 12 to 25 times more efficient than livestock at transforming low-protein feed into edible protein [10,72]. Much focus has recently been paid to evaluating the mealworm’s diet in order to enhance its nutritional properties. Research into the dietary requirements for mealworm development was initiated in the 1950s, when carbohydrate contents of approximately 80–85% were considered adequate for larval feed. However, extensive studies conducted over the following decades contradicted this claim, contributing to a better understanding of the real needs of mealworms during their larval phase. Today, the production of food for *T. molitor* larvae is based on the use of bran and other water-based raw materials, such as apples, carrots, or cabbage, as well as protein sources, including soy protein, brewer’s yeast, and casein [76]. Table 2 shows a summary of recent research on this topic.

The protein content of mealworms can be easily modified exogenously by manipulating the larval diet. In this regard, mealworms, as well as other insect species, have been found to efficiently utilize organic waste for growth, allowing waste management for a wide range of human activities, including agriculture, the food industry, and households. This approach is a cost-effective and environmentally friendly way to reuse leftover biological materials [77]. This transition from traditional bran-based diets may positively impact the physiological properties of larvae, resulting in faster growth and higher protein levels. In an experimental study, agricultural food waste was employed to feed mealworms. The results showed that using spent barley grains instead of other similar waste produces better larval growth, increasing biomass by 23.1% and reaching 24% DW protein. On the contrary, feeding mealworms with soybean pulp, known as okara, yielded 36% DW protein, 2% more than using oatmeal as a bran source, but resulting in a 30.4% decrease in biomass [78]. Another study obtained similar results when using spent grains to feed *T. molitor* larvae. The data displayed a good growth rate (7.5% per day) and an FCR (2.8) similar to that achieved with wheat bran (2.2), which was used as a control. Feeding larvae with this agri-food residue and wheat bran showed similar efficiency in the conversion of ingested food (ECIs) (weight gained/feed consumed) (36.4% vs. 46.7%, respectively). Low FCRs and high ECIs indicate efficient food use by mealworms [79]. Although wheat bran is considered a by-product of wheat flour, its substitution with other nutritious agri-food residues should be recommended due to the unfavorable methods involved during production [80].

Nitrogen-rich sources such as the agricultural by-products discussed above seem to be effectively utilized by mealworm larvae during growth and could be behind the improved yields achieved [81,82]. Jankauskienè et al. [83] identified the highest protein content in mealworms supplemented with spent brewer’s yeast grains (59.1% DW), suggesting a significant influence of dietary protein levels on the amount of protein accumulated in the larvae. Similarly, feeding efficiency might also be positively correlated with the protein content, as well as hemicellulose levels in the feeding. On the contrary, lignin content may exert the opposite effect [84].

Fermentation processes might positively affect the length and weight of larvae by supposedly improving the nutrient profile of feed. Akeed et al. [85] reported a significant (*p* < 0.05) increase in mealworm growth rates by fermenting cotton cake, the by-product of cottonseed processing, with the bacterial strain *Aspergillus tubingensis* FSS117. Larval weight was higher when larvae were fed with the previously fermented biological materials than with wheat bran (63.6 > 56 mg). The authors hypothesized the possible elimination of the polyphenol gossypol during the fermentation process, reducing the concentration of this toxic polyphenol and, consequently, larval mortality. In addition, fermentation also enabled the formation of various enzymes capable of increasing the protein content and improving the amino acid profile in the larvae’s diet.

Despite the apparent benefits reported by different studies on waste recycling for mealworm development, the reality seems to be more challenging. Using nutrient-dense agricultural waste to feed insects could impact livestock farming, as the two would compete for the same food source. This scenario would generate competition for feed, ultimately forcing farmers to use more commercial grains to achieve the same growth efficiency and leading to higher global feed prices. However, simply feeding insects commercially could also be counterproductive, since soy is commonly used as a feed ingredient for insect breeding. Due to soy production being a highly land-intensive activity, increasing the number of cultivable hectares to meet potentially higher demand would be environmentally harmful and clearly unsustainable. Using nutritious and widely distributed local plant species in specific areas of the globe instead of more traditional plant sources could be a more effective and sustainable option for feeding *T. molitor* larvae. The prickly pear fruit, and in particular the remaining spiny residue from processing, called cladode, was found to provide excellent results as a feeding material in mealworm farming. Errico et al. [86] studied the replacement of carrot and potato in the larvae’s diet with prickly pear cladodes, reporting an increase in larval growth (64.8 > 53.8–62.5 mg per larva). In addition, insects preferred the cladodes to the carrot and potato matrices, also showing higher overall qualities and longer shelf life. Therefore, prickly pear cladodes can provide a nutrient solution for mealworms instead of traditional plant sources, while also repurposing an underutilized biological material. This would help to significantly reduce the ecological footprint of raising *T. molitor* larvae.

The use of municipal waste streams for feeding insects can lead to the loss of sustainable energy inputs. Although insects would allow for more efficient use of energy, competition with the bioenergy sector would generate distrust regarding the environmental impact of this practice [87].

On the other hand, there are regulatory policies regarding the type of food allowed for insect feeding that prohibit the use of certain waste products depending on their origin. For instance, jurisdictions in the EU and the United Kingdom (UK) do not allow the use of waste that includes animal products, such as consumer food residues, slaughterhouse products, manure, human wastewater, and household waste, due to the potential presence of harmful end products. According to the regulatory policies of the EU, the only waste substrates considered safe for feeding insects intended for human consumption are those derived from processing residues and former foodstuffs of vegetable, dairy, egg, and/or honey origin (Regulation (EU) 2022/1104) [87].

Another challenge for using waste in mealworm feeding is the difficulty in obtaining a constant nutritional composition over time and space [87]. Variations in the organic matter composition may limit the amount of nutrients in the larval diet, which means a possible loss of quality in the insect development and protein production. In addition, the lack of standardized protocols for insect feeding makes comparison among studies difficult, affecting feed efficiency during breeding [88]. In this sense, international projects that involve a wide range of participants, including not only researchers but also companies and even public entities, may contribute to disseminating standardized rearing protocols due to the greater visibility of their activities. This is the case for the ValuSect Project, an Interreg North-West Europe-funded initiative dedicated to acquiring and spreading knowledge on the insect supply chain, from breeders to consumers. Due to the large number of participants conducting insect rearing experiments, a single feeding protocol was standardized for use by all partners involved. A similar strategy was adopted when feeding mealworms using wheat brans as dry feed. An international consortium, including companies working with this insect, was assembled and encouraged to participate in standardizing a feeding protocol for *T. molitor* larvae [89]. These attempts at unifying criteria can be extended to the use of waste side streams in insect feed, always in line with the regulatory measures of each region.

In relation to the organic waste side streams eventually used in insect feeding, they are exposed to various sources of contaminants, such as bacteria, mycotoxins, heavy metals, and pesticides, which potentially serve as disease vectors. Therefore, new strategies are needed to improve food safety, including the introduction of improved traceability of raw materials [87,90]. The sector’s rapid progress has gone beyond legislative frameworks, prompting global food safety authorities to ensure the safe production and processing of insects [91]. Reducing the contaminant load in insect substrates could be a viable measure. Although there is a lack of research on mealworm production, the study conducted on black soldier fly larvae suggests some advantages, but also difficulties when decontaminating the rearing substrate. For instance, larval activity itself may be able to reduce the microbial load in the gut or secrete antimicrobial compounds. Substrate pretreatment, such as milling, disintegration, thermal, radiological, chemical, and biological treatments, can also be considered as decontamination options. Heat pretreatment can be used to reduce microbial load in organic waste, but an increase in the levels of phenolic compounds and tannins, which are toxic to black soldier fly larvae, has been suggested. Irradiation of food waste before feeding effectively inactivated the original microbiota in these larvae but decreased production efficiency. Little research has been conducted so far, so there is an urgent need to expand knowledge on this topic to produce safe insect ingredients from agri-food side streams.

In an effort to improve insect protein yield, recently investigated strategies have focused on manipulating microbiomes to modify critical production traits. By applying genome editing, breeding, phage therapy, and biotics, host–microbiome interactions could be altered, enabling greater production efficiency and biosecurity while reducing operating costs. This approach ultimately offers a more sustainable and competitive insect rearing prospect for the future [92].

**Table 2 foods-14-04068-t002:** Recent research on feeding mealworms with agricultural by-products for sustainable production.

Food	Survival Rate (%)	Larvae Weight (mg)	FCR	Larvae Protein (% DM)	Reference
Oatmeal (control)	91.8	–	–	34	Yu et al. [78]
Okara	92.3	–	–	36	
Barley spent grain	98.3	–	–	24	
Sesame oil meal	91.7	–	–	23	
Spent coffee grounds	64	–	–	32	
Wheat bran (control)	92	107.8	2.2	–	Vrontaki et al. [79]
Sunflower by-product	0.7	109.8	36.4	–	
Lucerne by-product	55.3	73.5	8	–	
Oat by-product	79.7	102.9	3.2	–	
Spent grains	85	107	2.8	–	
Maize by–product	82.7	88.5	6.8	–	
Mill residues	47.7	18.8	39.8	–	
Rice bran	83	120.2	2.6	–	
Rice husk	43.7	30.2	10.7	–	
Spent mushroom substrate	13	6.9	146.8	–	
Wheat bran (control)	93.4	27.3	3.3	55.4	Musembi et al. [93]
Wheat bran and potato waste (75:25)	93.1	40.9	2.3	50	
Wheat bran and potato waste (50:50)	93.8	39.8	2.2	48.3	
Wheat bran and potato waste (25:75)	93.4	38.7	1.9	47.8	
Potato waste	92.5	22.9	2.1	43.3	
Oatmeal and apple (90:10) (control)	75–80	105	4.1	45.2	Bendowski et al. [94]
Oatmeal and plant by-products (75:25)	60	144	2.8	44.5	
Oatmeal and plant by-products (50:50)	85–90	165	3.3	44.9	
Oatmeal and plant by-products (25:75)	65–70	161	4	45.7	
Oatmeal and meat by-products (75:25)	80–85	179	3.1	51.5	
Oatmeal and meat by-products (50:25)	70–75	210	3.8	52.7	
Wheat bran (control)	91.6	68	–	38.3	Al–Mekhlaf et al. [95]
Wheat bran and bread leftovers (75:25)	90.8	92–96	–	38.7	
Wheat bran and bread leftovers (50:50)	89.6	104–108	–	28.7	
Wheat bran and bread leftovers (25:75)	94.8	96–100	–	25.7	
Bread leftovers	88.8	28	–	21.7	
Wheat bran (control)	96	–	11.8	44.5	Ferri et al. [96]
Wheat bran and chestnut shell (87.5:12.5)	98	–	8.1	52	
Wheat bran and chestnut shell (75:25)	98–99	–	7.9	46.2	
Corn flour (control)	96.7	8.9	–	34.8	Gulsunoglu–Konuskan et al. [97]
Corn flour and pomegranate peel (50:50)	98.7	10.5	–	27.6	

FCR: feed conversion ratio; DM: dry matter.

### 3.3. Applicability of Mealworm Protein in the Food Industry

Leaving aside the potential risks to human health posed by eating mealworms, which are discussed in other sections, their introduction into the human diet as a new source of high-quality protein represents a significant step toward the acceptance of insects as a sustainable way to maintain an adequate intake of this macronutrient in future society [98]. Since its approval by the EU for marketing as a dry matrix (whole or powder) (Commission Implementing Regulation (EU) 2021/882 of 1 June 2021) [99], *T. molitor* larvae have demonstrated a promising aptitude as a novel ingredient in the production of a diverse range of foodstuffs. Their physicochemical characteristics and nutritional properties were tested in recent research, which is discussed in the next paragraphs. Figure 3 shows different food products with the addition of mealworm flour.

#### 3.3.1. Bakery Products

Mealworm powder, also named mealworm flour due to its appearance, has been recently studied as an ingredient in the elaboration of novel breads and bakery products. The conventional ingredients used in breadmaking and other related products are considered not nutritious enough, as they are composed mainly of carbohydrates. Those produced with refined wheat flour are good examples of staple foods with low nutritional value. Therefore, the inclusion of edible insect flours in the recipe could enhance the quantity and quality of protein, as well as fat and dietary fiber [100]. Jankauskienè et al. [101] reported higher protein contents in wheat bread with different concentrations of mealworm flour (5%, 10%, and 15%), reaching values over 8.3 g/100 g of bread (control group) (9.4–12.3 g/100 g of bread). These increased protein levels affected the amino acid profile as expected, increasing the amount of EAAs, such as lysine, methionine, threonine, and tryptophan. On the other hand, not only did the addition of larvae flour influence the nutritional composition of the bread, but also the rearing conditions applied. This fact highlights the importance of diet in larval composition, as explored in the previous section. In contrast, despite the good protein profile achieved, the authors showed concern about some potentially problematic issues related to the addition of the insect protein. The porosity of the dough changed as a function of the amount and form of the insect protein used (milled and freeze-dried), as well as the feed used during rearing, likely caused by differences in the structure, solubility, and hydration of the proteins. Mealworm protein can also hinder dough formation during fermentation by weakening the gluten network, leading to poor gas retention. Still, the impact of this possible circumstance in the experiment remained unclear. However, Carpentieri et al. [102] produced durum wheat pasta with improved technological characteristics using mealworm flour (0–30%). They reported altered protein features, highlighting the formation of amylose–lipid complexes and hydrogen and electrostatic interactions between proteins and polysaccharides, which meant better dough stability and bioactivity.

The use of mealworm flour in other recent investigations also resulted in increased protein, significantly affecting the FA profile of the developed products. Kowaslki et al. [103] achieved 1.6 and 3.7 g more protein per 100 g of product using 15 and 30% larvae in a sponge cake produced with wheat flour, respectively. This inclusion also increased the monounsaturated FA (MUFA) content in the final product (47–48 > 43.2% of total FAs) but decreased the PUFA content (15.3–16.7 < 22.3% of total FAs), finally resulting in an unbalanced *n*6/*n*3 ratio (>6). Values considered healthy are ≤5 [104]. In another study, Draszanowska et al. [105] found the opposite when developing oatmeal cookies. In this case, the addition of mealworm flour significantly (*p* < 0.05) enhanced the value for this ratio (14.8–17.8 < 23.1), although it was still far from the recommended values. The MUFA portion increased, while PUFAs showed values comparable to the control group. On the other hand, protein content was also improved by 2–7 g/100 g of product.

Despite the nutritional profile achievements reported, the inclusion of mealworm flour in traditional bakery recipes remarkably affected sensory properties, leading to potential rejection by consumers when certain addition percentages are exceeded. Several examples of this circumstance are given in the literature [106,107,108]. Gantner et al. [109] noted a clearly different sensory profile in breads with more than 5% added mealworm flour. Higher concentrations broke the balance between all the sensory properties tested, resulting in a final product that was inappropriate for marketing. The addition of the insect flour produced darker bread doughs, a less typical bready odor and flavor, and greater bitterness and nuttiness. Similarly, Jankauskienè et al. [101] found a decrease in sensory quality scores as higher percentages of the insect flour were added to the bread, darkening the dough and negatively altering the texture and taste. This fact could seriously limit consumer acceptance.

#### 3.3.2. Meat Products

The partial substitution of meat proteins with those of insects is a growing research practice aimed at making better use of livestock raw materials by requiring lower amounts in meat processing. This target entails the challenge of achieving a proper integration between these two dissimilar matrices. In this sense, the most recent studies in the matter focused on developing sausages with physicochemical characteristics as close as possible to those of the original product. Among other tests, mealworm proteins were assessed for their ability to perform proper emulsions, which is essential in this type of product. At this point, various studies specified different limits for the addition of larvae. Zhang et al. [110] reported a lower emulsifying capacity when more than 5% mealworm flour was used in the production of pork frankfurters. As the replacement rate increased, the fluid released also increased due to the weaker interaction between insect and animal proteins. The increased fat release, also observed at higher concentrations, was attributed to the lower emulsifying ability of larval proteins compared to meat proteins. However, other aspects related to consumer acceptance, such as the interior color of the sausage, flavor, uniformity, and juiciness, presented a different aggregation limit (10%). A similar finding was reported in another study, in which the addition of mealworms to a fermented dry sausage could exceed 5% without negatively affecting the sensory perception of the panelists, who set this percentage as the maximum limit for product acceptance [111]. The maximum concentration found for mealworms and other edible insects to produce acceptable meat products should not exceed 10%. Higher percentages of inclusion would promote undesirable changes according to product standards [112].

The addition of mealworm flour into the meat products has been shown to potentially increase the protein content and improve the amino acid profile. The addition of 10% to pork hams resulted in 14% more protein, going from 35% to 49% DW. In the same way, mealworm protein increased the levels of almost all amino acids in the final meat product, although not in significant (*p* > 0.05) amounts, except for proline and tyrosine. According to the authors of the study, the increased protein levels were directly associated with the loss of moisture due to the lower water-binding capacity of the insect flour compared to traditional meat proteins. Still, the remaining final protein content is difficult to predict, since several crucial molecular events, such as protein interactions and structural rearrangements, as well as changes in textural properties, could happen in the meat matrix [113].

#### 3.3.3. Dairy Products

Inoculation of insect proteins into dairy products based on milky matrices was found to be compatible, reinforcing the protein content but significantly affecting sensory attributes, especially color. Zielinska et al. [114] reported higher protein amounts in the samples with insect proteins than in the control sample (from 1.9 to 3.1% with a 10% substitution) in a reformulated ice cream. This study showed a slower melting rate as the concentration of mealworm flour increased, hypothesizing an insect protein effect on this feature. Similarly, at the highest mealworm levels, the overrun was the lowest, probably increasing viscosity. According to the authors, the increased protein and the presence of chitin molecules might be behind this event. Higher overruns are preferable up to a point because they mean more volume and therefore more profits. However, overly low volumes result in a dense cream that consumers often dislike [115]. Still, Zielinska et al. [114] observed better physical properties for 10% mealworm substitution, also improving nutritional and nutraceutical aspects.

Yogurts are another dairy product with a potentially suitable matrix for containing mealworm flour. A recently conducted study displayed that its incorporation into yogurt formulations is feasible, increasing the protein content and improving other important characteristics. In this study, Andrzejczyk et al. [116] managed to incorporate freeze-dried powder of *T. molitor* larvae into a mixture of milk and skimmed milk powder to obtain a protein-fortified fermented product (from 4.9 to 7.4% using 5% larvae powder) with a higher FA content (2.2% more). The novel reformulated yogurt reached a faster curd formation and fermentation process, producing lower acidity matrices with lower firmness and consistency. The addition of mealworms clearly impacted whey syneresis, having possible repercussions for consumer acceptance. In addition, the darker color and bitter taste acquired by the yogurts were perceived as negative by participating panelists.

#### 3.3.4. Other Food Products

In addition to the food matrices discussed above, there are other less-researched food products that may be incorporated with mealworm flour to replace ingredients such as meat or simply to improve the nutritional profile. In this regard, mealworms were reported as a suitable ingredient to reformulate a traditional cream-type soap made of vegetables. When mealworm flour was added to the formulation of this product (20%), the panelists of the study, who comprised a large group of 100 people of different age ranges, rated it higher than creams made with different insects, including the house cricket, the lesser mealworm (buffalo worm), and the grasshopper (*Ruspolia differens*). The score reached a value of 6.6 points on a 10-point scale [117]. Similarly, the addition of minced mealworm to a pasta sauce, replacing 50% of the minced meat (pork and beef), showed comparable acceptance to sauces made entirely of minced meat (scores of 6.9 and 6.5 out of 9, respectively) [118].

The inclusion of mealworms in semi-liquid textured foods appears to provide good characteristics in the final product. García–Fontanals et al. [119] found that adding larva flour to a spreadable cheese in combination with a vegetable protein source, such as faba beans, could improve the quality of the product. Both flours showed a synergistic effect in improving texture by increasing firmness and stickiness and decreasing spreadability. The flavor was also enhanced, scoring higher than the control cheese. Specifically, the hybrid sample (with added insect and faba flours) developed a more intense cheesy flavor than a commercial plant-based reference. In addition, cheeses with mealworm flour were more flavorful than those without. Finally, the combination of milk protein and insect and faba flours displayed a higher protein content and lower saturated fat, starch, and sugar contents than commercial analogues. Ma et al. [120] reported that by adding 5% mealworm protein to a soy yogurt, the rheological properties can be modified, yielding a softer texture, similar to that of milk yogurt. The product reformulation also exhibited enhanced water retention and reduced syneresis, suggesting improved storage stability. In addition, the mealworm protein raised the level of this macronutrient in the yogurt, as well as those of free amino acids. Phenolic compounds also increased, boosting antioxidant capacity. The addition of mealworm proteins provided unique nutty and umami notes to the soy yogurt while maintaining a balanced sourness.

It can be inferred that the incorporation of mealworms into food products seems a promising approach to achieve increased protein matrices. However, the level of inclusion is one of the main barriers in the formulation of these products for now, since the addition of mealworm flours significantly affects consumer acceptance of the product by altering sensory properties. Furthermore, technological properties may also be damaged. On the other hand, improving labelling could have positive effects on reducing insect rejection through the selection of appropriate messages [121].

**Figure 3 foods-14-04068-f003:**
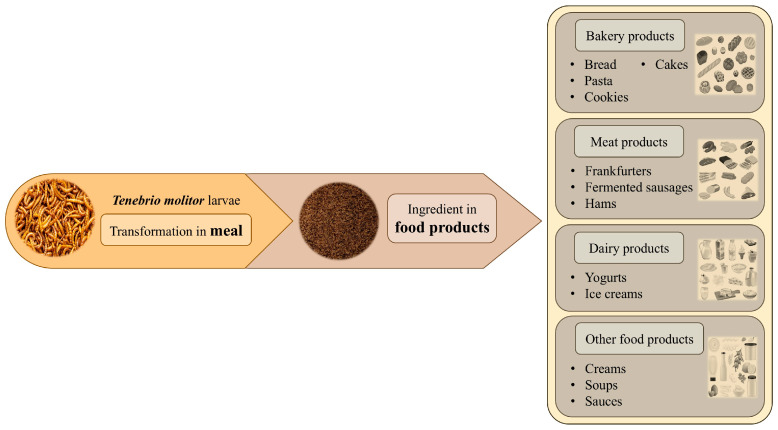
Potential incorporation of mealworm flour into food products, according to recent research discussed in the review.

### 3.4. Potential Adverse Health Effects of Eating Mealworms

According to recent research discussed in previous sections, mealworms are a promising substrate for food applications for several reasons: they have a good nutritional profile, require less energy to raise, and have a more efficient FCR compared to livestock. In addition, rearing mealworms releases lower GHG emissions. Apart from this, the mealworm matrix shows certain adequacy for use in the formulation of food products in the form of flours. These positive findings make the *T. molitor* larvae a potential sustainable source of novel proteins. Nevertheless, some potential negative aspects about their safety must be addressed. Despite the positive evaluation of the European Food Safety Authority (EFSA) for the marketing authorization of mealworms in the EU (Commission Implementing Regulation (EU) 2022/169) [65], the consumption of this insect can pose some health risks that should be addressed, since it is still very unknown in the food market. The exhaustive work carried out by the scientific community to determine the potential hazards derived from the consumption of *T. molitor* larvae, including the presence of allergens, foodborne pathogens, prions, and toxic substances, is summarized in Figure 4.

#### 3.4.1. Presence of Allergens

Most of the protein allergens identified in edible insects are widely distributed in nature. Specifically, their presence has been reported in other arthropods, such as dust mites, crustaceans, and mollusks. Nevertheless, despite the diversity and complexity of the insect protein network, the number of allergens known is very scarce. New proteins with allergenic properties are expected to be progressively discovered, thus increasing our knowledge about the insect allergens [122]. In this sense, the analysis of serum from allergic patients after ingesting insects or simply after being in contact with them may contribute to identifying these types of proteins. Under this approach, Beaumont et al. [123] identified the proteins responsible for severe food anaphylaxis induced by mealworms in a French male consumer. Proteomic analysis of his serum allowed the researchers to study the proteins responsible for the immune reaction. The presence of the larval cuticle proteins A1A and A2B and the hexamerin and tropomyosin epitopes not involved in mite or crustacean cross-reactivity was confirmed. In addition, the results showed possible sensitivity to the protein α-amylase, which has three-dimensional structures and sequences similar to those of the house mite *Dermatophagoides pteronyssinus* and to the protein tubulin, a potential pan-allergen.

In order to find the greatest number of mealworm proteins with allergenic properties, researchers suggested the study of possible cross-reactions with crustaceans or mite proteins. Van Broekhoven et al. [124] found cross-reactivity when sera from patients allergic to these two arthropods came into contact with the larvae proteins tropomyosin or α-amylase, hexamerin 1B precursor, and muscle myosin. This immunological reaction is based on the mistaken recognition of a protein by the body as an allergen due to its chemical similarity with another protein to which the body is sensitized. For this reason, the immune system of those consumers allergic to shellfish and/or mite proteins may also react to mealworm proteins. Palmer et al. [125] also found cross-reactivity in patients allergic to shrimp after consuming mealworm and other insect species. They reported weaker binding with the *T. molitor* larval tropomyosin than with the shrimp tropomyosin. This means a possible lower risk of allergic reaction when consuming mealworms compared to shrimp.

The possibility of reducing the allergenicity of edible insects by applying different processing methods was recently suggested [126]. Van Broekhoven et al. [124] observed a decrease in tropomyosin IgE cross-reactivity after heating mealworm proteins. In fact, post-harvest processing methods, such as heating (e.g., pasteurization and sterilization) or blanching, can also affect the allergenicity of proteins. In this regard, Lamberti et al. [127] reported the possibility of partially reducing the cross-allergenicity by heat-treating mealworms. However, some proteins involved in this adverse health phenomenon are heat-stable, so allergic patients should be cautious when consuming insects. On the other hand, the decrease of immunological reactions in sensitive consumers is not predictable, and the opposite might occur [128]. Thus, Broekman et al. [129] did not find a significant difference in the immune response of patients allergic to shrimp administered with heat-treated and non-heat-treated *T. molitor* larval protein extracts. In fact, 2 of the 15 patients used in the study showed an increase in allergic reaction after the thermal process. This was attributed to the greater solubility in PBS (phosphate-buffered saline) solution used as an extraction buffer for some allergens.

Studies conducted to determine allergenicity in insects reported that some proteins are not only capable of triggering immune reactions, but also the carbohydrate chitin. Both this polysaccharide and chitinase, the enzyme responsible for digesting chitin and also present in the insect’s body, can induce an immune response [130]. Chitin is found mainly in the lungs or intestine, where it activates a series of innate (eosinophils, macrophages) and adaptive immune cells (IL-4/IL-13 expressing T helper type-2 lymphocytes). In addition, chitin induces cytokine production, leukocyte recruitment, and alternative macrophage activation. Immune recognition of chitin also involves pattern recognition receptors, mainly through TLR-2 and Dectin-1, which activate immune cells and induce the production of cytokines and the creation of an immune network, ultimately resulting in inflammatory and allergic responses [131]. The possibility of treating mealworms to reduce the allergenicity caused by this oligosaccharide was already addressed in a previous section.

In summary, those people allergic to dust mites and/or shellfish should take extreme precautions if they wish to include mealworms in their diet, since they run the risk of suffering a cross-reaction. On the other hand, consumers sensitive to the carbohydrate chitin could suffer the same fate. In this regard, companies marketing products made of insects, such as mealworms, should provide appropriate labeling warning about the potential risk of developing allergic reactions [122]. In this regard, Commission Implementing Regulation (EU) 2022/169 recommends showing warnings near the list of ingredients about possible allergy risks for people allergic to shellfish and related products and dust mites [65].

#### 3.4.2. Potential Presence of Pathogenic Microorganisms

Some anatomical parts of the insect’s body, such as the gut, are hosts for microorganisms and their toxins. The diversity of the microorganism community depends on a number of different factors, including vertical transmission from mother to offspring, rearing conditions, and processing [132]. This microbial load can persist throughout the entire production chain and reach the consumer, potentially causing a public health problem.

In an attempt to shed light on the microbiological risk of consuming mealworm, Costa et al. [42] analyzed the microbial content once the breeding stage was completed and the insect size was suitable for sale. These authors observed that starvation and refrigeration for 8 days resulted in a significant reduction of the initial microbial load, including not only bacteria but also yeasts and molds. Specifically, the total viable aerobic count (TVAC) was reduced from 7.7 to 7.1 log colony-forming units (CFU)/g and Enterobacteriaceae from 6.7 to 6.3 log CFU/g. Nevertheless, this decrease was not sufficient to reduce the microbial load below the legal limits for ground meat or raw materials used in meat preparation, whose limits are 6.7 and 3 log CFU/g for TVAC and Enterobacteriaceae, respectively. On the other hand, the presence of the pathogens *Escherichia coli*, *Staphylococcus* coagulase-positive, *Listeria monocytogenes*, and *Salmonella* spp. was not detected, revealing appropriate hygienic practices throughout the production chain. Conversely, Mohammadsalim et al. [133] reported the finding of pathogens, such as *Bacillus cereus*, *Pseudomonas aeruginosa*, *Enterobacter asburiae*, *Bacillus firmus*, *Serratia marcescens*, and *Staphylococcus* sp. In this case, the presence of pathogenic microbes might be related to less-strict hygienic handling measures, since the larvae were acquired in a local market.

To understand to what extent accidental contamination with pathogens can pose a real hazard to the health of mealworm consumers, some authors assessed the persistence capacity of pathogens in the larvae and the effectiveness of different elimination procedures. In this regard, Mancini et al. [73] intentionally contaminated the rearing substrate (brewery spent grains and bread) with *L. monocitogenes* and found that the pathogen survived the entire process without affecting larval survival. Washing the insects with sterile saline solution did not produce any significant effect on the microbial load. In contrast, food deprivation for a period of 24–48 h reduced the content of *L. monocitogenes* by around 2 log CFU/g. On the other hand, cooking at 150 °C for 10 min fully removed the pathogen from the larvae.

In accordance with Commission Implementing Regulation (EU) 2022/169 [65], the microbiological criteria for the sale of powdered or dried mealworm limit the microbial load to the levels indicated in Table 3. Therefore, the implementation of processing methods to reduce or prevent the risks of microbial contamination in *T. molitor* larvae is essential to meet the standard requirements requested by the European laws.

Kooh et al. [134] proposed a hazard analysis critical control plan (HACCP) for the production of mealworm powder with the aim of limiting the risk to consumer health. These authors identified and analyzed the main critical points of the different stages of the manufacturing process, which included the microbiological hazards related to the presence of certain expected microorganisms, such as *B. cereus*, *Clostridium botulinum*, *Clostridium perfringens*, *L. monocytogenes*, *Staphylococcus aureus*, *Campylobacter* spp., *Cronobacter* spp., *Salmonella* spp., and *Yersinia* spp. In addition, the possible presence of shiga toxin-producing *E. coli* (STEC), as well as the hepatitis A virus and human norovirus, the main viral cause of acute gastroenteritis in most countries of the world [135], was also considered. The application of HACCP also covered the process of making burgers, protein shakes, baby porridge, and biscuits with mealworm flour. The results shown in this study might be very useful to establish a quantitative microbial risk assessment system [134].

A positive aspect of mealworms is their potential ability to metabolize mycotoxins produced by accidental contamination of the growth substrate with molds. Guo et al. [136] reported non-mortality after consumption of wheat grains colonized by the molds *Fusarium proliferatum* and *Fusarium poae*, both producers of fumonisins, enniatins, and beauvericin. However, high levels of the mycotoxin enniatin were found to affect larval survival. Van Broekhoven et al. [124] observed that mealworms can excrete 14% less deoxynivalenol compared to the total consumed in the diet, and 41% less when the feed was intentionally contaminated, thus suggesting a detoxifying power. Mealworm can also tolerate levels of aflatoxin B1 higher than 0.4 mg/kg of dry feed, an amount approximately 20 times higher than the legal limit established for the EU, which is 0.02 mg/kg. Despite this possible metabolic advantage of the mealworms, Commission Implementing Regulation (EU) 2022/169 [65] established maximum limits for mycotoxins (Table 4) due to their health hazard as potential promoters of kidney, neurological, and cancer diseases [137].

Regarding the threat of possible transmission of viruses from mealworms to consumers, studies carried out to date have not demonstrated that viruses belonging to the *Iridovirus* and *Densovirus* genera, which are capable of infecting the *T. molitor larva* [138,139], can be transmitted to humans [140]. In fact, both bacteria and insect pathogenic viruses are harmless to humans due to the phylogenetic differences between these living beings.

Mealworms may serve as a vehicle for many foodborne pathogens, which could eventually lead to outbreaks. Therefore, maintaining high-quality hygiene measures, avoiding contamination with microorganisms during breeding, and ensuring adequate control throughout the production stages, including distribution, should be implemented in an analogous way to that of any other animal production, to guarantee food safety.

**Table 3 foods-14-04068-t003:** Legal limits for microorganisms in *Tenebrio molitor* larvae established by Commission Implementing Regulation (EU) 2022/169 [65].

Microorganism	Maximum Count for Frozen Mealworms (log CFU/g)	Maximum Count for Powdered or Dried Mealworms (log CFU/g)
Total aerobic colony count	≤5	≤5
*Enterobacteriaceae (presumptive)*	≤2	≤2
*Escherichia coli*	≤1.7	≤1.7
*Listeria monocitogenes*	Absence in 25 g	Absence in 25 g
*Salmonella* spp.	Absence in 25 g	Absence in 25 g
*Bacillus cereus (presumptive)*	≤2	≤2
*Staphylococcus* coagulase-positive	≤2	≤2
Sulfite-reducing anaerobes	≤1.5	≤1.5
Yeasts and molds	≤2	≤2

EU: European Union; CFU: colony formation units.

#### 3.4.3. Risk of Prion Transmission

The risk of spreading transmissible spongiform encephalopathies (TSEs) or animal prion diseases has long been treated with indifference, considering that the zoonotic potential of these diseases was negligible. This was due to the natural barrier that limits the propagation of prions from one species to another, as well as the lack of epidemiologic evidence of an association with human prion diseases. However, the emergence of a new human prion disease caused by a variant of Creutzfeldt–Jakob disease (vCJD), which triggered an unprecedented public health crisis in Europe, led to a significant change in the attitude of health authorities [141].

Today, epidemiological surveillance programs have revealed the incidence and prevalence of prion diseases in the ruminant population that until now had been unnoticed. Transmission experiments in non-human primates and transgenic mice clearly indicate that some types of prions can have the potential to infect humans. Therefore, it is essential to take extreme measures towards these potentially zoonotic agents [141].

Insects, including mealworms, are being investigated as potential transmitters of prions. These pathogens are naturally misfolded proteins that accumulate in the brain [142]. Some insect species can act as mechanical vectors for these proteins when they are present in the rearing substrate. In addition, these prions would be highly resistant to variable environmental conditions, maintaining their infectious capacity in soil and water for long periods of time [143]. However, exhaustive control of the substrate greatly reduces the probability that prions can reach the consumer [144]. Therefore, preventive measures and the prohibition of using substrates of human and ruminant origin might prevent the entry of prions into the food chain [143]. Furthermore, the use of plant substrates for insect breeding further reduces the eventual presence of prions [145].

#### 3.4.4. Potential Contamination with Hazardous Organic Compounds

The use of certain chemical substances, such as pesticides on farmland, has caused the accumulation of biological contaminants in the environment with detrimental effects on human health. These organic compounds can be present in large portions of edible plant residues, and these residues can be used as a substrate in insect rearing. These chemicals can not only reach the insects’ bodies, but also bioaccumulate during their life cycle, and subsequently be transferred to the human body through consumption [146].

Among the harmful chemical compounds that can reach the consumer, dioxins and dioxin-like polychlorinated biphenyls (PCBs) are especially concerning, as they have been involved in incidents occurring within the food chain and during feed production [147]. Therefore, it would be advisable to investigate to what extent new foods such as insects, especially the mealworm, may serve as a vehicle for these organochlorine compounds. A recent study showed bioaccumulation of PCBs of 77–82% when a contaminated substrate (wheat bran) was used at a concentration of up to 24.4 ppb. This level of contamination did not affect the survival of the larvae. On the other hand, drying the product notably increased bioaccumulation to a maximum of 233%. These figures were expressed as follows:BAF (bioaccumulation factor) (%) = 100 × concentration of PCBs in larvae/concentration of PCBs in wheat bran.

This indicates a clear and dangerous ability of the mealworm to accumulate PCBs from the diet [148]. Interestingly, Ratel et al. [149] did not observe bioaccumulation of this organochlorine. However, its presence was confirmed in the mealworms studied after feeding them contaminated wheat bran. Furthermore, these authors also highlighted the significant effect of desiccation on the concentration of PCBs in the larval body, increasing by almost three times. Desiccation is a mandatory process to produce mealworm powder.

On the other hand, dioxins were found in *T. molitor* larvae at levels below the limit established by the EU for these chlorinated compounds in fish (0.0001 and 0.25 < 5 pg World Health Organization (WHO) toxic equivalents (TEQs)/g [150]. Table 4 shows the maximum concentrations of dioxins and PCBs permitted by the EU in mealworms reared for human consumption.

These investigations demonstrate the importance of considering organochlorine chemical contaminants in the quality analysis of mealworms intended for both human consumption and animal feed.

#### 3.4.5. Possible Presence of Heavy Metals

When we talk about heavy metals, we refer to those that have a high density, specifically five times more than water, and a large atomic mass, and whose presence in the environment might have harmful consequences, as well as their ingestion [151,152].

This group of toxic and relatively accessible metallic elements, which include zinc, copper, arsenic, mercury, and lead, could reach insects such as mealworms through contamination of breeding feed. Truzzi et al. [153] showed a significant correlation between the mercury content in the feeding substrate of *T. molitor* larvae and the worm itself, indicating bioaccumulation according to the BAF (concentration in the organism/concentration in the feed) reported by the authors, which increased from 1.5 to 6.9. Despite this, the values obtained in the study for mercury were lower (0.1 × 10^−^^3^ and 0.5 × 10^−3^ mg/kg) than those of other heavy metals also identified, including cadmium (0.008–0.02 mg/kg), lead (0.06–0.08 mg/kg), nickel (0.03–0.6 mg/kg), arsenic (~0.02 mg/kg), and selenium (0.06–0.09 mg/kg).

The treatment of the larval growth substrate with pesticides leaves residues of heavy metals, such as cadmium, lead, mercury, and arsenic. Likewise, the use of animal manure and other agricultural waste contributes to the deposition of these elements in the insect’s body [154]. Regarding this, Noyens et al. [155] observed the presence of heavy metals after feeding mealworms with different organic side streams. Specifically, higher amounts of lead and chrome were found in the larvae fed with fermented chicory roots. In addition, these metals showed bioaccumulation by presenting BAFs of 2.7 and 1.1 for lead and chrome, respectively. Still, the reported levels of heavy metals in the mealworm were low and not dangerous to health. On the other hand, the analysis of the different feed substrates studied showed heavy metal contents below the standard limits established by the EU (0.4 mg/100 g dry matter (DM) and 0.04 mg/100 g DM for lead and cadmium, respectively). Similarly, the reported contents of these two heavy metals in mealworms fed with wheat bran, lentil flour, and both were between 0.05 and 0.08 mg/kg DM for cadmium and between 0.05 and 0.07 mg/kg DM for lead [156]. The values were below the legal limit established by the EU for dried larvae (≤0.075 mg/kg for lead and ≤0.1 mg/kg for cadmium) [65].

It is important to note that certain heavy metals, such as copper, zinc, or manganese, are essential for the proper functioning of the body, belonging to the group of minerals known as oligoelements or trace elements. Nevertheless, excessive intake of these metals, exceeding the recommended limits, may have harmful consequences. Others, including lead, mercury, and arsenic, can even be harmful at low concentrations. These elements are considered systemic toxicants and occupy the top positions on the list of hazardous substances [157].

The absence of heavy metals in *T. molitor* larvae is definitely not guaranteed, and low levels of these elements can be found. Therefore, larvae should be subjected to exhaustive controls to guarantee compliance with the limits established by the authorities. As discussed above, heavy metals may reach the larvae through diet. This necessitates the implementation of effective detection protocols in the rearing substrate, thus preventing the transfer of these toxic compounds to the insect during growth. This measure will undoubtedly contribute to improving food safety.

**Table 4 foods-14-04068-t004:** Legal limits for contaminants in *Tenebrio molitor* larvae established by Commission Implementing Regulation (EU) 2022/169 [65].

Contaminant	Maximum Quantity in Frozen Mealworms	Maximum Quantity in Powdered or Dried Mealworms
Heavy metals		
Lead (mg/kg)	≤0.01	≤0.075
Cadmium (mg/kg)	≤0.05	≤0.1
*Mycotoxins*		
Aflatoxins (Sum of B1, B2, G1, and G2) (µg/kg)	≤4	≤4
Aflatoxin B1 (µg/kg)	≤2	≤2
Deoxynivalenol (µg/kg)	≤200	≤200
Ochratoxin A (µg/kg)	≤1	≤1
Dioxins and PCBs		
Sum of dioxins and dl-PCBs (UB, WHO-TEQ2005) (**) (pg/g fat)	≤0.75	≤0.75

EU: European Union; ** Sum of polychlorinated dibenzo-para-dioxins (PCDDs), polychlorinated dibenzofurans (PCDFs), and dioxin-like polychlorinated biphenyls (PCBs) expressed as World Health Organization toxic equivalent factors (using WHO-TEFs of 2005).

**Figure 4 foods-14-04068-f004:**
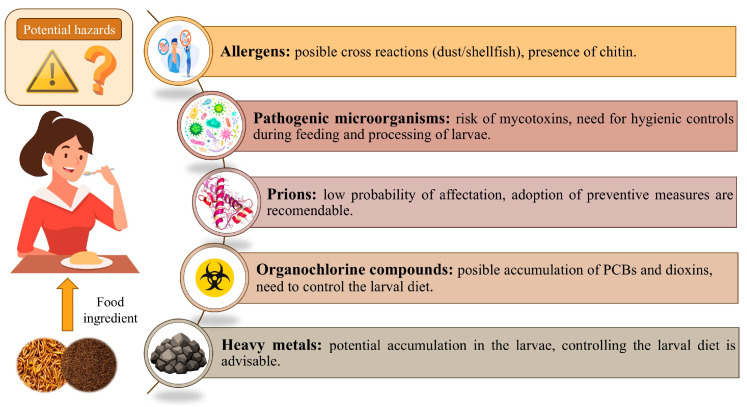
Potential health risks of incorporating *Tenebrio molitor larvae* into food products. PCB: polychlorinated biphenyl.

### 3.5. Consumer Acceptance of Mealworm

The EU is strongly committed to sustainability in the medium–long term, promoting better action in terms of emissions, land, and water use. In this regard, the Farm-to-Fork strategy, which was designed at the heart of the European Green Deal to address climate and environmental-related challenges, aims to outline a new sustainable food system, linking health with a balanced environment [158]. Insect farming fits this target by supporting healthy diets and supporting the transition toward a more sustainable future.

Despite being a nutritious and sustainable alternative to meat protein, the acceptance of insects by Western society may represent a significant barrier to their commercialization. However, the consumption of insects is a very widespread practice in many cultures, such as in Asian countries, where insects are included in the regular diet, as well as in some communities in South America. Specifically, in Latin America, entomophagy is a widespread eating habit that is still part of its culinary heritage [19,159]. This significant difference between cultures around the world is driven by their attitude toward insect consumption, which is highly influenced by inherited eating habits. Thus, Northern Europeans show a greater disposition toward insect consumption than other nearby Western communities, such as Central Europeans. Of course, product shape is also fundamental in insect acceptance studies. Germans can tolerate better the aspect of insect-based products than the visible, whole insect’s body. However, Chinese consumers do not differentiate between the two forms. Thailand is another Asian country where eating insects is not an unfamiliar practice, being highly appreciated for their taste. In contrast, in South Korea, there is no trend toward eating insects, although entomophagy is steadily increasing. Indians refuse to include insects in their diets for cultural reasons, as otherwise they would violate their own moral code [160]. African culture is exceptionally attached to insects, having consumed them for generations, to the point that 90% of diets on this continent are supplemented with edible insects [161].

Therefore, it might be concluded that consumer acceptance is closely related to many aspects, such as familiarity, preparation, insect visibility, or neophobia, that are widely related to the culinary heritage of every part of the globe. An interesting recent study analyzing early adopters’ willingness to accept frozen, ready-to-cook mealworms showed a greater inclination among people from countries with a tradition of eating insects. The authors of the study indicated the desire for new experiences and ease of preparation as key factors for incorporating mealworms into the diet [162].

The sensory evaluation, along with the willingness to consume and to buy products made of mealworms, showed that the addition of insects in a non-visible way is better accepted than using entire insects. A greater preference for salty foods with added mealworms than for sweet foods was observed, demonstrating the importance of an appropriate combination [22]. Tzompa–Sosa et al. [163] found healthiness to be the primary driver of acceptance of whole and processed mealworms across a survey conducted in different countries, including Belgium, China, Italy, Mexico, and the United States; meanwhile, “aversion” or “dislike” were the most important factors in rejecting them. Italian consumers often showed a higher aversion to eating insects compared to consumers in the other countries participating in the study, giving less importance to health-related aspects. Information about the advantages of consuming mealworms in terms of sustainability and nutrition, and regarding food safety regulations, impacted more on consumers from Mexico, a country with entomophagy traditions. In contrast, the age of Chinese consumers had a greater influence on acceptance. In general, people under 42 years old from countries without entomophagy traditions showed more interest in processed mealworms than older people [164]. This outcome was similar to that reported by other authors who found that mealworms and other edible insects added to a cream soup might be less accepted by older people, although the data obtained were not significantly conclusive. Sensory parameters, such as texture and taste, had the most influence on overall product acceptance [117]. Ju et al. [165] applied different sacrificing methods for mealworms to assess consumer acceptability by conducting testing panels. Appearance, flavor, texture, and overall acceptance were significantly modified depending on the method used, demonstrating that alternatives to increase consumer acceptance of insects are possible. When mealworms were sacrificed using freezing and sonication, acceptance scores were reduced. In contrast, the use of a heating process, such as blanching or roasting, can increase consumer acceptance, as well as physicochemical properties. In this way, a dry heating and cooking procedure after sacrifice produced a strong roasted odor and tasty flavor. The use of traditional ingredients, such as spices and other natural ingredients, during food production might help mask the flavor of mealworms, adding other, more recognizable flavors. Mazurec et al. [166] reported limited insect aroma perception when making muffins using mealworms and other insects, suggesting possible masking by other ingredients (e.g., eggs, milk, or canola oil). On the other hand, fat is primarily responsible for both pleasant and unpleasant aromas in food matrices. In this sense, Bologna-type sausages with added defatted mealworm powder obtained a better score than sausages added with non-defatted mealworm powder. This fact was attributed to the extraction of fat using supercritical CO_2_, which removes volatile compounds, reducing the intensity of the mealworm powder odor [167].

A study conducted in South Korea identified sensory properties (e.g., taste, color, and odor) as the characteristics most valued by consumers highly interested in insect-based foods. Other key factors for these types of consumers were nutritional value, cost, and food safety [160].

Providing the benefits of a healthy or sustainable diet may positively modify consumer tolerance to mealworm products by increasing the sensory threshold to taste and the number of insects added [168]. A recent systematic study about marketing measures for insect-based products identified several decisive purchasing aspects. Insect-containing foods should be marketed emphasizing the presence of insects and/or highlighting the benefits of their high protein content. On the other hand, to enhance the taste and the perception of quality and safety, it is necessary to influence socially, promote the product through advertising, and carry out tasting activities [169].

Price is another key factor that determines the willingness to pay for a food product, as it plays a prominent role in consumer acceptance and purchasing decisions. The price of food may act in two different and contradictory ways. High prices might lead to product rejection by consumers, while low prices might have the opposite effect. However, higher prices could also be interpreted as a quality attribute and increase the consumer purchase intention [170]. The results of a behavioral experiment on how prices may contribute to purchase expectations of a mealworm burger showed that a higher price was indeed related to product quality, positively influencing willingness to pay. Therefore, it can be inferred that higher initial prices could increase purchases and even establish a stable demand for these types of products in Western countries [171].

Consumer acceptance of mealworm products definitely depends on many different factors that intervene in the perception of this new food ingredient. On the other hand, it would be interesting to carry out experiments in real-life environments to assess consumer behavior towards these products, thus identifying potential buyers [172].

### 3.6. Economic Viability of Mealworm Farming for Human Consumption

Edible insects can be produced by three different methods: wild harvesting, which accounts for 92% of the total insect supply, semi-domestication, which represents 6%, and the remaining 2% of insects intended for human consumption are farmed. Insect farming is the most recent way of producing edible insects, especially in developed countries [173].

Overuse of cropland threatens the survival of edible insects. Pesticides have increased insect mortality worldwide, diminishing invertebrate populations by up to 67% in the last 40 years. In addition, uncontrolled overharvesting is causing changes in the trophic chain, and the greater demand for new insect species is leading to a change in classic harvesting techniques for others more aggressive. On the other hand, semi-domestication helped reduce environmental and health risks caused by wild harvesting of insects, increasing sustainability and saving time. This way of producing insects involves partial control over breeding, mortality, space use, and food supply. All of this carries potential economic benefits. Nevertheless, full domestication through farming is currently the best system for providing insects, as most species can be easily raised in small spaces, have short life cycles, and agricultural waste can be used as feed. Moreover, insect farming can be developed in both urban and rural areas, and short-term benefits may be feasible [174].

There are good opportunities in East Africa for breeding a variety of edible insects, with potentially higher benefits than livestock and crop production. In the last few years, insect farming has increased dramatically in this part of the world due to the low cost of rearing and easy access to organic waste. In countries such as Kenya, Tanzania, and Uganda, several companies have emerged, operating as microenterprises, with the potential to evolve toward more efficient and automated systems. Today, there is still little information on the costs and profitability of edible insects in Africa, but the price of insect meal is higher than that of fishmeal in Europe. This is expected to reverse as farms adopt a circular and more sustainable economy [175].

Although Latin America is a region with an entomophagous tradition and the second-largest market for edible insects worldwide, most urbanized areas reject insects in their diet. In addition, the insect-based industry is underdeveloped, and there are hardly any strategies to boost and spread the business. On the other hand, there is a lack of research and training institutions, which makes it difficult to finance new insect-based companies. Mass production of insects might provide innovative products more cheaply than the edible insects currently marketed in the region [176].

Insect farming is a growing business in Europe, with several companies currently operating in different countries. Most of these companies are micro- and small-sized companies, accounting for 42 and 60% of the total, respectively. The rest are considered medium-sized companies. While micro and small-sized companies barely reach 10 employees, the latter can have 50–250 employees. The total investment of many of these companies is below EUR 500,000, while around 30% is between EUR 1 and 5 million, and 6% above EUR 10 million. Unfortunately, the reality is that many of these European companies are struggling to survive and need private capital and/or public financing. This sector is still navigating through a business model that copes with several challenges, such as high production costs, unsteady demands, and low revenues. Regulations can also be a headache for these young companies, creating uncertainty and hindering the achievement of profitability [142,177].

Niyonsaba et al. [177] assessed the robustness, which refers to the technical and economic feasibility, of different business models for insect production under different future scenarios. Regarding the mealworm, the authors identified a cooperative-based business model as appropriate according to recent bibliographic information from experts. Specifically, mealworms would be processed for food production using multiple rearing sites but just one processing facility. The idea of the business model is based on the rearer managing the breeding operations, with minimal mechanization, and the cooperative organizing the processing and sanitation stages. This business prototype is designed to produce an intermediate product that can be used as an ingredient in the formulation of mealworm-based food products. The study identified a series of stress factors or uncertainties that can constrain the profitability of the planned business model. Despite the decline in secondary stream prices due to liberal regulations, their low levels for mealworm feed make them insufficiently profitable. Therefore, experts in the field suggest that stricter regulation might benefit mealworm production, supporting this claim with the quality and safety of the final product obtained. On the other hand, the low degree of mechanization mentioned above raised the hypothesis of a slight increase in energy costs. In addition, transportation costs can be reduced by the proximity of the side stream source.

Niyonsaba et al. [178] analyzed the profitability of a quarter of the mealworm farming sector in the Netherlands, inquiring the investment required to develop such activity, including facilities, machinery, administration, marketing and quality, rentals, and variable expenses, such as feed and utilities (e.g., energy, gas, and water). Similarly, revenues were also assessed, represented mainly by sales of fresh larvae. Four of the seven farms investigated displayed a negative net present value, which suggests unprofitability in most mealworm farms. Increased viability should be based on operating in small market segments, which allow for higher prices. Furthermore, improving consumer acceptance of insects and their derivatives could help increase demand. According to the authors, additional revenue sources, such as offering extension services, could be applied. Certainly, due to the instability of demand and the small size of the insect market, large-capital investments could pose an unaffordable risk.

Commercial insect breeding is booming in industrialized areas of the world, such as North America and Europe, and although it may currently be considered a profitable business in the medium–long term, it is not yet an economically convenient activity in general terms, especially for small-scale production, due to the high price of the raw materials used as substrate. However, it is predictable that as technology enhances and is incorporated into a circular economic scenario, production costs will decrease, increasing business profitability. Furthermore, policy interventions that subsidize mechanization and encourage standardization of procedures, such as insect breeding and final quality, could have a positive impact on profitability. Promoting insects as food might also stimulate consumer demand [179].

## 4. Conclusions

The high pressure that livestock farming exerts on the ecosystem, combined with the incessant increase in demand for meat, suggests a change of direction based on the introduction of alternative sources of protein into the diet. Breeding insects, particularly mealworms, might help alleviate pressure on natural resources and reduce the GHG released to the atmosphere due to their unique physiological characteristics. Furthermore, their high protein content and good overall nutritional profile, including significant amounts of PUFAs, such as omega-3 and omega-6, would contribute to better nourishment and greater food security in monetarily depressed societies. In addition, current insect feeding methods are attempting to look toward a circular economic scenario by using organic wastes while maintaining or even improving efficiency. However, there is still little data available, and several challenges must be addressed, such as the competition for food substrate with livestock, regulatory policies, and the maintenance of feed quality over time, among others.

The use of mealworms in the food industry has led to the production of new protein-enriched products, but with serious technological and sensory deficiencies when the level of inclusion exceeds 5 or 10%, negatively affecting their acceptance by consumers. Further research is urgently needed to overcome the complex problem of developing mealworm-based food products by designing new methods for obtaining and incorporating the insect powder. However, some health matters might hinder the definitive introduction of these products into the food market. Possible cross-reactivity with crustacean or mite proteins and the carbohydrate chitin would harm the health of sensitive consumers by inducing immune responses. Poor hygienic processing practices could lead to pathogen contamination and promote a high microbial load in the larval body. Similarly, irresponsible production practices might favor the presence of organochlorine contaminants, such as PCBs and dioxins. However, by applying strict hygiene and risk control standards, food safety should be guaranteed. Adequate warning about consuming mealworm-based products might prevent serious health complications.

Finally, the negative perception of insects by consumers, especially in Western countries, is a strong barrier to their expansion as an alternative source of protein. Therefore, it is important to find well-received recipes based on experimentation, but also to promote the positive aspects of insect consumption, such as the benefits of following a healthy and sustainable diet. Nevertheless, social influence cannot change the reality of insect farming, which is still an emerging business with low profitability prospects, as some recent studies suggest. Still, regulatory policies might help generate greater rentability by subsidizing mechanization and boosting the standardization of production procedures. The insect business is expected to be profitable in the medium–long term as technology improves and is incorporated into a circular economic scenario, reducing costs and increasing profits.

## Figures and Tables

**Figure 1 foods-14-04068-f001:**
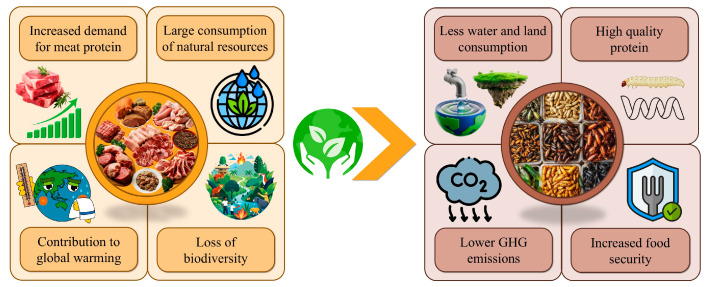
Potential advantages of insects for a sustainable protein supply compared to meat proteins. GHG: greenhouse gas.

## Data Availability

All data and information shared in this review study are available for consultation in the Google Scholar and Scopus databases.

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
