# Peer review of "Feasibility of Using Tenebrio molitor Larvae as an Alternative Protein Source"

_foods, 2025, doi:10.3390/foods14234068_

Round 1

Reviewer 1 Report

Comments and Suggestions for Authors

This review systematically evaluates the feasibility of yellow mealworm (Tenebrio molitor) as an alternative protein source, covering nutritional value, farming practices, food applications, potential health risks, consumer acceptance, and economic feasibility. The manuscript is clearly structured, rich in data, and provides substantial reference value.

Major Issues and Suggestions:

  1. While the review compiles extensive data, it lacks deeper critical analysis of methodological limitations, conflicting conclusions, and gaps in current knowledge. Suggestion: Add a brief “research gaps” or “future directions” subsection at the end of each major part. Where data are controversial (e.g., protein digestibility, fatty acid ratios, allergen stability), provide comparative discussion and highlight methodological differences. 
  2. Although Asian entomophagy traditions are mentioned, the consumer acceptance section remains dominated by European and North American contexts, with insufficient coverage of Africa, Latin America, and South-East Asia. Suggestion: Expand the discussion to include consumer behavior, cultural preferences, and market potential in non-Western regions. The use of cross-cultural consumer psychology or behavioral economics frameworks would add depth. 
  3. While the manuscript mentions insect farming activities in Europe, Africa, and Latin America, it does not adequately address cost structure, profitability, policy incentives, or technological barriers. Suggestion: Include specific case studies (e.g., a European insect farm) with cost–benefit data, and analyze how policies (such as EU Novel Food regulations) influence market entry. 
  4. The manuscript notes that processing (e.g., heating, fermentation) affects nutrition and allergenicity, but lacks a systematic summary. Suggestion: Add a comparative table/figure showing how different processing methods (e.g., drying, freezing, fermentation, ultrafine grinding) impact proteins, fats, vitamins, allergens, and microbes. Discuss the balance between “minimal processing” and “functional enhancement.” 
  5. Some figures (e.g., Figures 2, 3, 4) contain dense information and have brief captions, making them less reader-friendly. Suggestion: Redesign or split complex figures, highlight key points, and expand captions to specify data sources, comparison objects, and main conclusions.

The paper makes a valuable contribution and, after revision, could serve as a representative review in the field.

Comments on the Quality of English Language

English is generally clear. Minor polishing is recommended.

Author Response

Thank you very much for your comments and suggestions. You will find our responses below.

- Comment 1: while the review compiles extensive data, it lacks deeper critical analysis of methodological limitations, conflicting conclusions, and gaps in current knowledge. Suggestion: Add a brief “research gaps” or “future directions” subsection at the end of each major part. Where data are controversial (e.g., protein digestibility, fatty acid ratios, allergen stability), provide comparative discussion and highlight methodological differences.

Response: sections “3.2.1. Protein content”, “3.2.2. Other nutritional compounds of mealworm, and “3.5.1. Presence of allergens”, were expanded to comply with your suggestion.

- Comment 2: although Asian entomophagy traditions are mentioned, the consumer acceptance section remains dominated by European and North American contexts, with insufficient coverage of Africa, Latin America, and South-East Asia. Suggestion: Expand the discussion to include consumer behavior, cultural preferences, and market potential in non-Western regions. The use of cross-cultural consumer psychology or behavioral economics frameworks would add depth.

Response: according to the literature, the most problematic introduction of insect-based products occurs in the Western market. However, this section can be improved by adding information about all that you mention, so based on your suggestions, the section “3.6. Consumer Acceptance of Mealworms” was expanded.

- Comment 3: while the manuscript mentions insect farming activities in Europe, Africa, and Latin America, it does not adequately address cost structure, profitability, policy incentives, or technological barriers. Suggestion: Include specific case studies (e.g., a European insect farm) with cost–benefit data, and analyze how policies (such as EU Novel Food regulations) influence market entry.

Response: section “3.7. Economic viability of mealworm farming for human consumption”, which addresses this topic, was expanded to include your suggestion.

- Comment 4: the manuscript notes that processing (e.g., heating, fermentation) affects nutrition and allergenicity, but lacks a systematic summary. Suggestion: Add a comparative table/figure showing how different processing methods (e.g., drying, freezing, fermentation, ultrafine grinding) impact proteins, fats, vitamins, allergens, and microbes. Discuss the balance between “minimal processing” and “functional enhancement.”

Response: when we discussed different processing methods in the manuscript, we referred to different aspects related to the mealworm. For instance, fermentation is mentioned when discussing different ways to improve mealworm yield during feeding. On the other hand, the heating process of the mealworm was related to the digestibility of its protein or the impact on allergenicity as you mention in your comment. Therefore, when processing methods are discussed, they refer to different topics within the same overall subject. We would have to establish a discussion and comparison in several sections, and we believe this could deviate the focus of the review. We would probably need to extend too much the length of the manuscript as it is a very broad issue.

- Comment 5: some figures (e.g., Figures 2, 3, 4) contain dense information and have brief captions, making them less reader-friendly. Suggestion: Redesign or split complex figures, highlight key points, and expand captions to specify data sources, comparison objects, and main conclusions.

Response: the figures 2, 3, and 4 were modified as you recommend improving readability. The corresponding captions were also rewritten to clarify the explanation of the figures.

Reviewer 2 Report

Comments and Suggestions for Authors

This manuscript presents a comprehensive and timely review on the potential of Tenebrio molitor (mealworm) as a sustainable alternative to conventional animal protein. The topic is highly relevant to the scope of Foods, addressing critical issues of food security, environmental sustainability, and novel food sources. The review is well-structured, logically organized, and covers a broad range of essential aspects, including nutritional profile, farming practices, food application, health risks, consumer acceptance, and economic viability. The authors have done an excellent job in synthesizing a significant body of recent literature, and the data presentation in tables and figures is a particular strength.

The manuscript is generally well-written and provides a balanced perspective, discussing both the advantages and the challenges of mealworm utilization. After addressing the points outlined below, I believe this manuscript will be a valuable contribution to the field and suitable for publication in Foods.

Abstract and Conclusion Sections:

Abstract: The current abstract is somewhat descriptive and reads like a table of contents. It should be condensed and restructured to provide a more powerful summary. It should clearly state the background/problem, the aim of the review, the key findings (e.g., nutritional comparability to meat, primary challenges in safety and acceptability), and the main conclusion (e.g., that mealworms are a promising alternative, but their feasibility hinges on overcoming technical, regulatory, and social hurdles).

Conclusion (Section 4): While the conclusion summarizes the paper adequately, it could be more impactful. It should be distilled to highlight the most critical take-home messages and provide more forward-looking, specific recommendations for future research and industry development, rather than primarily recapping previous sections.

Figures and Tables: The manuscript refers to Figures 1-4 and Tables 3-4, but their content is not provided in this draft. For the final submission, all figures must be included with high quality and clarity, and all tables must be complete.

Deepening of Discussion:

Section 3.3 (Farming): The discussion on using organic side streams is excellent. To strengthen it, consider adding a brief discussion on potential strategies to mitigate the inherent challenges, such as standardization protocols for waste substrates or decontamination processes to ensure safety and consistent nutrient profiles.

Section 3.4 (Food Application): When discussing the negative impact on sensory properties, it would be valuable to briefly mention ongoing strategies to mitigate these issues, such as the use of defatted insect meal, flavor masking techniques, or specific processing methods (e.g., extrusion) that can improve texture and flavor.

Section 3.5.1 (Allergens): The text presents conflicting findings on the effect of thermal processing on allergenicity. This section would benefit from a more conclusive summary sentence that clearly states the current uncertainty and its implication for risk management (e.g., "Given the current contradictory evidence, caution and clear labeling remain prudent for protecting susceptible consumers.").

Terminology Consistency: Ensure consistent use of terms throughout the manuscript. For instance, the title uses "Tenebrio molitor larva," while the text often uses "mealworm" and "T. molitor larvae." It is recommended to choose one primary term (e.g., "mealworm larvae") and use it consistently.

Abbreviations: Define all abbreviations upon first use. For example, FCR (L248), ECI (L277), DIAAS (L160), etc., should be spelled out when they first appear.

Data Presentation:

L215-216: The units for vitamin content ("between 100 and 1000 mg/g DL") seem improbably high (1000 mg/g = 100%). Please verify the original sources, as this is likely a unit error (e.g., it should be µg/g or mg/100g).

Table 1: The superscript letters for Zinc (Zn) values (e.g., 96.5c, 114.1b) are not explained in the table footnote. Please define these or remove them if they are irrelevant.

L23: "re–educate the poor perception" → See major comment 2.

L33: "non–stopping increasing demand" → See major comment 2.

L215-216: "all between 100 and 1000 mg/g DL" → Please check the units. This appears to be an error.

L248: "(feed consumed/weight gained)" → This is the inverse of the standard FCR calculation (weight gained/feed consumed). Please verify and correct the definition.

L576: "Commission Implementing Regulation (EU) 2022/169 concluded that..." → This is a very important regulatory point. It would be helpful to briefly state why the EU deemed warnings unnecessary despite the evidence (e.g., due to the very low number of reported cases).

Comments on the Quality of English Language

The manuscript requires thorough proofreading to correct minor grammatical errors and improve phrasing for clarity and conciseness. Examples include:

L23: "re–educate the poor perception" → "improve the negative perception" or "address the poor perception".

L33: "non–stopping increasing demand" → "continuously growing demand".

L57-58: "releases fewer greenhouse gases (GHGs) and less NH4 (ammonium), NH3 (ammonia), and CH4 (me-thane)" → This could be rephrased for precision, e.g., "releases fewer greenhouse gases (GHGs) and produces lower emissions of ammonia and methane".

References must be formatted consistently according to the journal's guidelines. Please ensure that all in-text citations (e.g., use of "et al.") and the reference list are uniform.

Author Response

Thank you very much for your comments and suggestions. You will find our responses below.

- Comment 1: abstract: the current abstract is somewhat descriptive and reads like a table of contents. It should be condensed and restructured to provide a more powerful summary. It should clearly state the background/problem, the aim of the review, the key findings (e.g., nutritional comparability to meat, primary challenges in safety and acceptability), and the main conclusion (e.g., that mealworms are a promising alternative, but their feasibility hinges on overcoming technical, regulatory, and social hurdles).

Response: as you comment, the abstract is deficit as it does not adequately reflect the important aspects of the manuscript, so it was completely rewritten following your advice.

- Comment 2: conclusion (section 4): while the conclusion summarizes the paper adequately, it could be more impactful. It should be distilled to highlight the most critical take-home messages and provide more forward-looking, specific recommendations for future research and industry development, rather than primarily recapping previous sections.

Response: the conclusion section was extensively modified to reflect your recommendation.

- Comment 3: figures and tables: the manuscript refers to Figures 1-4 and Tables 3-4, but their content is not provided in this draft. For the final submission, all figures must be included with high quality and clarity, and all tables must be complete.

Response: We apologize for the confusion. The tables and figures presented were modified several times during the preparation of the manuscript, and the final versions do not match their references in the text. We have corrected this mistake throughout the manuscript and added tables 3 and 4.

- Comment 4: section 3.3 (farming): the discussion on using organic side streams is excellent. To strengthen it, consider adding a brief discussion on potential strategies to mitigate the inherent challenges, such as standardization protocols for waste substrates or decontamination processes to ensure safety and consistent nutrient profiles.

Response: based on your suggestion, this section was improved by expanding the discussion on the topics you mention.

- Comment 4: section 3.4 (food application): when discussing the negative impact on sensory properties, it would be valuable to briefly mention ongoing strategies to mitigate these issues, such as the use of defatted insect meal, flavor masking techniques, or specific processing methods (e.g., extrusion) that can improve texture and flavor.

Response: the manuscript was improving by adding this topic to section “3.6. Consumer acceptance of Mealworm” because we though it fits better in this part of the manuscript.

- Comment 5: section 3.5.1 (allergens): The text presents conflicting findings on the effect of thermal processing on allergenicity. This section would benefit from a more conclusive summary sentence that clearly states the current uncertainty and its implication for risk management (e.g., "Given the current contradictory evidence, caution and clear labeling remain prudent for protecting susceptible consumers.").

Response: at the end of this section, we conclude that consumers can be assuming risks by consuming mealworm-based products, and the EU regulatory policies consider appropriate to show in labels the potential risk of allergic reactions, although it is not mandatory From “In summary”…to “dust mites”, the last paragraph of the section, which was modified during the peer review.

- Comment 6: terminology consistency: ensure consistent use of terms throughout the manuscript. For instance, the title uses "Tenebrio molitor larva," while the text often uses "mealworm" and " molitor larvae." It is recommended to choose one primary term (e.g., "mealworm larvae") and use it consistently.

Response: the manuscript was revised entirely looking for inconsistences. Regarding the example you mention, in the introduction we explained the following “from this point in the text we will refer to this insect as mealworm and T. molitor alternately), attempting to make the text more dynamic. We decided to use both terms interchangeably.

- Comment 7: abbreviations: define all abbreviations upon first use. For example, FCR (L248), ECI (L277), DIAAS (L160), etc., should be spelled out when they first appear.

Response: all abbreviations throughout the manuscript were revised and corrected.

- Comment 8: L215-216: The units for vitamin content ("between 100 and 1000 mg/g DL") seem improbably high (1000 mg/g = 100%). Please verify the original sources, as this is likely a unit error (e.g., it should be µg/g or mg/100g).

Response: as you suggest, the units are incorrect. They are actually mg/kg. This was corrected in the manuscript and in Table 1.

- Comment 9: Table 1: the superscript letters for Zinc (Zn) values (e.g., 96.5c, 114.1b) are not explained in the table footnote. Please define these or remove them if they are irrelevant. Response: Response: This table was redone several times and the letters do not belong to this version, so they were deleted.

- Comment 10: L23: "re–educate the poor perception" → See major comment 2.

Response: the phrase was modified and the expression changed to “negative perception”.

- Comment 11: L33: "non–stopping increasing demand" → See major comment 2.

Response: this expression was removed and the phrase rewritten as “excessive consumption in recent decades and the projected increase by 2050”.

- Comment 12: L215-216: "all between 100 and 1000 mg/g DL" → Please check the units. This appears to be an error.

Response: the units are wrong. They are actually mg/kg. It was corrected in the manuscript.

- Comment 13: L248: "(feed consumed/weight gained)" → This is the inverse of the standard FCR calculation (weight gained/feed consumed). Please verify and correct the definition.

Response: this ratio is well-expressed. The Feed Conversion Ratio (FCR) is feed consumed/weight gained, and the Efficiency of conversion of ingested food (ECI) is “weight gained/feed consumed”.

Below are some references to corroborate this statement.

Vrontaki, M., Adamaki-Sotiraki, C., Rumbos, C. I., Anastasiadis, A., & Athanassiou, C. G. (2024). Valorization of local agricultural by-products as nutritional substrates for Tenebrio molitor larvae: A sustainable approach to alternative protein production. Environmental Science and Pollution Research, 31(24), 35760-35768.

Bordiean, A., Krzyżaniak, M., Aljewicz, M., & Stolarski, M. J. (2022). Influence of different diets on growth and nutritional composition of yellow mealworm. Foods, 11(19), 3075.

- Comment 14: L576: "Commission Implementing Regulation (EU) 2022/169 concluded that..." → This is a very important regulatory point. It would be helpful to briefly state why the EU deemed warnings unnecessary despite the evidence (e.g., due to the very low number of reported cases).

Response: this paragraph was modified after carefully reading the Regulation again. The text states that it is recommended that the risk of allergy should be indicated on the label if it contains frozen, dried, or powder mealworms.

- Comment 15: L23: "re–educate the poor perception" → "improve the negative perception" or "address the poor perception"

Response: This expression was changed to “negative perception”. The phrase was also modified.

- Comment 16: "non–stopping increasing demand" → "continuously growing demand".

Response: the phrase containing this expression was changed to “excessive consumption in recent decades and the projected increase by 2050”.

- Comment 17: L57-58: "releases fewer greenhouse gases (GHGs) and less NH4 (ammonium), NH3 (ammonia), and CH4 (me-thane)" → This could be rephrased for precision, e.g., "releases fewer greenhouse gases (GHGs) and produces lower emissions of ammonia and methane".

Response: The phrase was modified as you recommend.

- Comment 18: References must be formatted consistently according to the journal's guidelines. Please ensure that all in-text citations (e.g., use of "et al.") and the reference list are uniform.

Response: the manuscript was entirely revised and some errors were corrected.

Reviewer 3 Report

Comments and Suggestions for Authors

Feasibility of using Tenebrio molitor larva as an alternative food to animal protein

Feasibility of using Tenebrio molitor larva as an alternative supplement for both animal and human foods.

Given the amount of text devoted to using mealworm as a supplement to human diets I think it should get equal billing with animal feed supplements.

Agregán R, Echegaray N, Moraga–Babiano L, Pateiro M & Lorenzo JM

General

I think the title needs rewriting. Throughout the text there are lots of incidents where figures from the references are quoted to 2 decimal places. Is that justified in the paper???

The 1st sentence in the introduction is 6 lines in length and if you break it down it doesn’t say much. Why not delete the 1st para.

In addition, what is the purpose of mentioning the other insect species. They are only mentioned in one other part of the paper. Also some discussion is needed as too why the mealworm was selected in this review. Most importantly there are no advantages/disadvantages of working with the mealworm.

In the last para of the introduction the authors set up the aim of the review. It needs to be rewritten as it does not set out the questions that being asked (or should be asked in the review).

A problem throughout the review is that the number of decimal places changes from three to full digits. I would suggest that whole number in exceptional circumstances 2 decimal places. This needs to be addressed.

The review seems to fall short on a vision as to what can be expected in the short and longer term.

Abstract

Line 16 - the authors seem undecided whether the mealworm is seen as a supplement to animal feeds, or as a direct supplement to human diets.

Line 22 - “ breeding, incorporation to food products” to “breeding, production system, incorporation into food products”

Introduction

Lines 30 to 35 - this is an excessively long awkward sentence to start an introduction of a paper. Preferably it needs to be deleted.

Although meat provides essential nutrients for human body development such as proteins, vitamins, and minerals, also reducing nutritional deficiencies and promoting health, the excessive high consumption along the last few decades, and the non–stopping increasing demand until 2050, specially in low and middle–income countries, is depleting natural resources, since meat has significantly larger environmental and climate foot-prints than production of plant–based foods.

Line 35 - the logic does not flow. Why would mini farmed species address the sustainability issues

I would suggest that the 1st para of the introduction be deleted.

Line 51 - why not just have one name applied to a species. It is confusing otherwise to alternate between the latin and common name

Line 52 - what nutrients??

Line 60 - protein sources that can be used as a protein supplement for animal feeds

Line 63 - I am not sure that the word “deeply” is appropriate. I would delete this work.

Units for 9 and 6.5 are nor given? Are these mean sensory score???

Line 59/60 - this sentence needs to place the supplementation potential into context. Ie the mealworm can be used as a protein supplementation for animal feeds as as a direct component for products targeting consumers. I think the review needs to outline some of the questions being asked. Some questions may be solved using the literature and some may needs additional research. Most importantly the review should set the scene for the way forward. What research into this subject should be tackled in the short medium and long term.

Line 130 - no reference to the best time/developmental stage to harvest the mealworm to produce the best meal.

Li ne 131 - I think the paper would be better served by separating sections into “Animal Feed Supplements” and “Human Food Supplements”

Line 155 - I think a comment on the palatability of the supplement for both animal feeds and Human Supplements is required. If no references exist then surely that it a high priority to research this area??

Line 164 - I think that fat, carbohydrate and ash should be used as subheadings.

Line 166 -m why give whole digits for some references and 2 decimal places for others (which I contend is not justified)

Line 200 - the threshold for sweetness is 0.125% whilst the reported total sugars in mealworm was 0.120%. I would argue that it is not a stop/start threshold for sweetness so an effect would be seen starting to be seen.

Line 215 - does an allergenic reaction to chitin occur with rumen digestion. I suspect the bacteria would digest the mealworm into VFAs and NH3 and then bacteria

Line 228 - Cannot see the relevance of this paragraph. Why are the authors reviewing meat composition data?? I would delete.

Line 300 - I find the argument in this para strange. Surely if there was competition for a food resource then farmer would feed it to the system that had the highest profitability!!

Line 222 - I assume the same constraints do not apply to feeding animals waste products. This is where splitting it up into supplements for animals and human foods would be an advantage

Line 391 - but decreased the PUFA content (15.34–391 16.73<22.27%), Need to explain the units???, both here and elesewhere.

Line 469 - Interestingly the review covers palatability for incorporation into human food, but does nor comment on palatability as a supplement by animals?

Line 480 - The units for the figures 6.9 and 6.5 are not given? Are they mean sensory scores??? Are they out of 10 or 15 as the highest scale.

Line 504 - I would suggest that lack of consumer acceptance of a insect meal into the new product and most importantly the labelling of the product may be the limiting factor to inclusion of mealworm into new products

Lines 506 and lines 501 - why the similar headings for 3.5 and 3.5.1???

Line 906 - the subheading within Table 1 need attention. “Major composition” should be “Proximal Composition”.

At what stage in the lifecycle was proximal composition measured? Larvae or adult mealworm.

Do the authors think that 2 decimal places is necessary? Wouldn’t 1 or full digits do.

FA should be in full. AA and EAA etc should also be in full.

Line 910 - the tables and acronyms need to stand by themselves. Currently that is not the case

Comments on the Quality of English Language

English is a problem. Sentences are unnecessarily long and difficult to follow. More sub-heading would help and better use of paragrahing. 

Author Response

Thank you very much for your comments and suggestions. You will find our responses below.

- General comment.

I think the title needs rewriting. Throughout the text there are lots of incidents where figures from the references are quoted to 2 decimal places. Is that justified in the paper???

The 1st sentence in the introduction is 6 lines in length and if you break it down it doesn’t say much. Why not delete the 1st para.

In addition, what is the purpose of mentioning the other insect species. They are only mentioned in one other part of the paper. Also some discussion is needed as too why the mealworm was selected in this review. Most importantly there are no advantages/disadvantages of working with the mealworm.

In the last para of the introduction the authors set up the aim of the review. It needs to be rewritten as it does not set out the questions that being asked (or should be asked in the review).

A problem throughout the review is that the number of decimal places changes from three to full digits. I would suggest that whole number in exceptional circumstances 2 decimal places. This needs to be addressed.

The review seems to fall short on a vision as to what can be expected in the short and longer term.

Responses: the title was changed to “Feasibility of using Tenebrio molitor larvae as an alternative protein source”. We think it matches better with the aim of the review.

Regarding the use of two decimals places, we removed them and maintained only one. This was applied in the entire manuscript.

We believe the first paragraph is important for presenting the problem. However, it was improved and shortened.

We wanted to briefly present the problem arising from the excessive demand for meat and different current strategies to mitigate it. Then, when discussing the option of using mealworms, we though that it would be a good idea to present some of the most researched insects, which are allowed for human nutrition in the EU. As for why we chose to work with mealworm, it was because it is the most researched insect today, and it is in the spotlight. However, we added an explanation of our decision in the introduction.

On the other hand, the entire review is based on the advantages and disadvantages of using mealworms with respect to many aspects, such as those related to nutrition, breeding performance, the presence of toxic compounds, consumer acceptance, and even economic viability as a profitable business.

Finally, the aim of the review was improved as you suggest.

- Comment 1: abstract.

Line 16 - the authors seem undecided whether the mealworm is seen as a supplement to animal feeds, or as a direct supplement to human diets.

Line 22 - “ breeding, incorporation to food products” to “breeding, production system, incorporation into food products”.

Responses: The abstract was rewritten, and we believe that it now better reflects the background, objective, key findings, and future prospects.

- Comment 2: introduction.

Lines 30 to 35 - this is an excessively long awkward sentence to start an introduction of a paper. Preferably it needs to be deleted.

Response: We agree. It was rewritten.

- Comment 3: Line 35 - the logic does not flow. Why would mini farmed species address the sustainability issues.

Response: We agree, so the phrase was rewritten.

- Comment 4: Line 51 - why not just have one name applied to a species. It is confusing otherwise to alternate between the latin and common name

Response: as the review is a long type of scientific article and that word is used many times throughout the manuscript, we believe that using both expressions alternately gives dynamism to the text instead of using the same word all the time. However, if it is highly advisable to choose just one, we will do so.

- Comment 5: Line 52 - what nutrients??

Response: it is explained in the next phrase “Thus, the mealworm stands out for its high protein content, with a balanced amino acid profile, healthy fatty acids, vitamins, and minerals”.

- Comment 6: Line 60 - protein sources that can be used as a protein supplement for animal feeds.

Response: that is true, but the main objective of this review is to discuss the feasibility of the mealworm for human nutrition, exclusively.

- Comment 7: Line 63 - I am not sure that the word “deeply” is appropriate. I would delete this work.

Response: We agree, it was a spelling error. It was changed to “in depth”.

- - Comment 8: Units for 9 and 6.5 are nor given? Are these mean sensory score???

Response: we are not able to find these figures.

- Comment 9: Line 59/60 - this sentence needs to place the supplementation potential into context. Ie the mealworm can be used as a protein supplementation for animal feeds as as a direct component for products targeting consumers. I think the review needs to outline some of the questions being asked. Some questions may be solved using the literature and some may needs additional research. Most importantly the review should set the scene for the way forward. What research into this subject should be tackled in the short medium and long term.

Response: The review focuses on human nutrition, what regard to the use of mealworms as an ingredient in food development. We understand that the issue you expose is another important topic, but it should be addressed in a separate paper. Furthermore, the weaknesses of the topics raised and future perspectives were addressed throughout the manuscript according to our criteria and based on recent published research. However, additional text about this was added in several parts of the review, as follows:

  • Section 3.2.1. Protein content: from “However, certain limitations…” to “…are considered excellent”.
  • Section 3.2.2. Other nutritional compounds of mealworm: from “As can be observed…” to “…nutritional profile” and from “Despite the concern…” to “…considering commercialization”.
  • Section 3.3. Suitability of recent trends in mealworm farming for production yield: from “In addition…” to “…each region” and from “Little research…” to “…side streams”.
  • Section 3.7. Economic viability of mealworm farming for human consumption: from “This sector…” to “…achievement of profitability”, from “experts in the field…” to “…side stream source”, from “Increased viability…” to “…unaffordable risk”, and from “In addition…” to “…consumer demand”.
  • Conclusions: from “Further research…” to “…insect matrix” and from “Therefore, it is important…” to “…production procedures.

- Comment 10: Line 130 - no reference to the best time/developmental stage to harvest the mealworm to produce the best meal.

Response: This depends on many external factors, such as humidity, temperature, and diet. However, although there is no fixed breeding period or exact harvest time, we added a few quotes on this topic to this section (3.1. Taxonomy, origin, morphology, and lifecycle).

- Comment 11: Line 131 - I think the paper would be better served by separating sections into “Animal Feed Supplements” and “Human Food Supplements”

Response: The review article focuses on mealworms for human nutrition and their viability as an ingredient. What you propose is also a popular topic, but we consider that it should be addressed in another article as it is too broad to fit into this manuscript.

- Comment 12: Line 155 - I think a comment on the palatability of the supplement for both animal feeds and Human Supplements is required. If no references exist then surely that it a high priority to research this area??

Response: the sensory attributes of mealworm-based products were analyzed in several parts of the manuscript. Section “3.4. Applicability of mealworm protein in the food industry” addresses this topic in all subsections. In addition, “section 3.6. Consumer acceptance of mealworm” discusses this in depth. Regarding the sensory analysis of mealworm flour itself, we do not know if there are human studies, but the priority and primary focus of academia and the food industry is the development of food products, so we think that addressing the palatability of mealworms as a pure ingredient is not realistic.

- Comment 13. Line 164 - I think that fat, carbohydrate and ash should be used as subheadings.

Response: we agree. We added three extra subheadings: 3.2.2.1. Fat, 3.2.2.2. Carbohydrates, and 3.2.2.3. Vitamins and minerals.

- Comment 14: Line 166 -m why give whole digits for some references and 2 decimal places for others (which I contend is not justified).

Response: this was corrected according to a previous comment. Finally, except for a few specific cases, like established thresholds or limits, only one decimal place was used.

- Comment 15: Line 200 - the threshold for sweetness is 0.125% whilst the reported total sugars in mealworm was 0.120%. I would argue that it is not a stop/start threshold for sweetness so an effect would be seen starting to be seen.

Response: We agree with your opinion, so we corrected this in the manuscript.

- Comment 16: Line 215 - does an allergenic reaction to chitin occur with rumen digestion. I suspect the bacteria would digest the mealworm into VFAs and NH3 and then bacteria.

Response: I unknown this event, but the review is focused on human nutrition, so this part of the manuscript addressed the effect of chitin on human health.

- Comment 17: Line 228 - Cannot see the relevance of this paragraph. Why are the authors reviewing meat composition data?? I would delete.

Response: we attempted to compare, whenever possible, the nutrient content and quality of mealworms with those of meat. Meat is considered the main source of protein in the human diet. Therefore, if the purpose is to partially replace its consumption with mealworm-based products, we consider it interesting and valuable to establish a nutritional comparison.

- Comment 18: Line 300 - I find the argument in this para strange. Surely if there was competition for a food resource then farmer would feed it to the system that had the highest profitability!!

Response: what we mean by this statement is that if both a farmer who raises insects and a farmer who rear livestock use the same source of feed, there could be some competition for it, in this case agriculture waste.

- Comment 19: Line 222 - I assume the same constraints do not apply to feeding animals waste products. This is where splitting it up into supplements for animals and human foods would be an advantage

Response: we agree with your opinion. In this lines, what we wanted to transmit is that not using some waste from human activities to feed mealworms intended for human consumption could be a barrier for breeders, but at the same time, we consider these measures understandable, since they try to protect human health.

- Comment 20: Line 391 - but decreased the PUFA content (15.34–391 16.73<22.27%), Need to explain the units???, both here and elesewhere.

Response: it means % of total fatty acids, it was explained immediately before. However, we can understand some difficulties following the reading, so this issue was corrected in the entire manuscript.

- Comment 21: Line 469 - Interestingly the review covers palatability for incorporation into human food, but does nor comment on palatability as a supplement by animals?

Response: the article is focused on human nutrition. The topic is so huge that extending the review adding content related to animal feeding would be little approachable. This is a good idea for another review.

- Comment 22: Line 480 - The units for the figures 6.9 and 6.5 are not given? Are they mean sensory scores??? Are they out of 10 or 15 as the highest scale.

Response: They are sensory scores, but they may be difficult to understand, so we added a clarification to the text.

- Comment 23: Line 504 - I would suggest that lack of consumer acceptance of a insect meal into the new product and most importantly the labelling of the product may be the limiting factor to inclusion of mealworm into new products.

Response: The first part or your suggestion was already commented in this part of the manuscript, but we added an allusion to the labelling because it is also an important matter for accepting a food product as you mention, especially relevant in this case.

- Comment 24: Lines 506 and lines 501 - why the similar headings for 3.5 and 3.5.1???

Response: It was a mistake. Thanks for commenting. It was corrected. The real title is “3.5.1. Presence of allergens”.

- Comment 25: Line 906 - the subheading within Table 1 need attention. “Major composition” should be “Proximal Composition”.

Response: We agree. It was a mistake. “Major composition” was changed to “proximal composition”.

- Comment 26: At what stage in the lifecycle was proximal composition measured? Larvae or adult mealworm.

Response: it refers to the larval stage. Table 1 mentions that the data are expressed as dry larvae.

- Comment 27: Do the authors think that 2 decimal places is necessary? Wouldn’t 1 or full digits do.

Response: this question was answered in another previous comment and corrected in the manuscript, using only 1 decimal place for all figures.

- Comment 28: FA should be in full. AA and EAA etc should also be in full.

Response: We agree. All the abbreviations used in the manuscript were revised, including the ones you mention. Fatty acid was abbreviated.

- Comment 29: Line 910 - the tables and acronyms need to stand by themselves. Currently that is not the case

Response: we defined the acronyms used in all table footnotes. We revised this carefully.

Round 2

Reviewer 2 Report

Comments and Suggestions for Authors

After the manuscript was revised, its quality was greatly improved.

Author Response

We are pleased to know that our responses to your comments have been satisfactory and the changes made to the article have contributed to improving its scientific quality.